# Active learning-guided optimization of cell-free biosensors for lead testing in drinking water

Brenda M. Wang [1,14], Nicole Chiang[2,14], Holly M. Ekas[3,4,5], Dylan M. Brown [3,5], Garrett Dildine [6], Tyler J. Lucci[3,5], Siyuan Feng [5,7], Vanessa Bly [5,8], Jean-François Gaillard [6], Julius B. Lucks [3,5,9,10], Ashty S. Karim [3,4,5], Diwakar Shukla [2,11,12,13] & Michael C. Jewett [1,2,3,4,5] ✉

Point-of-use diagnostics based on allosteric transcription factors (aTFs) are promising tools for environmental monitoring and human health. However, biosensors relying on natural aTFs rarely exhibit the sensitivity and selectivity needed for real-world applications, and traditional directed evolution struggles to optimize multiple biosensor properties at once. To overcome these challenges, we develop a multi-objective, machine learning (ML)-guided cell-free gene expression workflow for engineering aTF-based biosensors. Our approach rapidly generates high-quality sequence-to-function data, which we transform into an augmented paired dataset to train an ML model using directional labels that capture how aTF mutations alter performance. We apply our workflow to engineer the aTF PbrR as a point-of-use diagnostic for lead contamination in water. We tune the sensitivity of PbrR to sense at the U.S. Environmental Protection Agency (EPA) action level for lead and modify the selectivity away from zinc, a common metal found in water supplies. Finally, we show that the engineered PbrR functions in freeze-dried cell-free reactions, enabling a diagnostic capable of detecting lead in drinking water down to ~5.7 ppb. Our ML-driven, multi-objective framework powered by directional tokens can generalize to other biosensors and proteins, accelerating the development of synthetic biology tools for biotechnology applications.

Bacteria constantly sense their environment to regulate gene expression and mitigate toxins. Many of these sense-and-respond systems rely on allosteric transcription factors (aTFs), which bind to specific DNA operator sequences and undergo conformational changes only when a cognate small-molecule inducer is present. This mechanism activates or represses downstream gene expression and has inspired the design of gene circuits using an aTF and its operator as biosensors for specific ligands[1]. Such biosensors have shown great promise as synthetic biology tools[2-5] and molecular diagnostics for environmental (e.g., pesticides[6], heavy metals[7], and contaminants[8,9]) and human health (e.g., disease markers[10], hormones[11]) monitoring[12-25].

Unfortunately, natural aTFs usually do not meet the performance requirements for real-world applications without engineering[2]. Properties such as sensitivity, selectivity, dynamic range, and response time often need improvement. High-throughput screening and directed evolution can address these limitations, but engineering aTFs remains difficult due to their allosteric nature[26,27] and the challenge of simultaneously tuning multiple parameters (e.g., sensitivity and selectivity)[28-30]. A key bottleneck is the ability to quickly map sequence–function relationships across multiple ligands, capturing both successes and failures, to inform forward engineering.

Machine learning (ML)-guided directed evolution has transformed the way we navigate vast protein sequence-function landscapes, making exploration faster and less reliant on exhaustive experiments[31–39]. At its core, ML supports protein engineering in two complementary ways: predictive models, which score given sequences or structures, and generative models, which propose new ones.

Predictive models act as evaluators to screen predefined libraries and prioritize candidates for experimental validation. Examples include zero-shot predictors, which infer fitness directly from evolutionary or structural context using pretrained protein large language models (pLLMs)[40–46], and supervised models trained on sequence-function datasets to guide optimization for specific tasks such as catalysis[43,47–50]. Classification-based models, like DeepTFactor[51] (which identifies transcription factors) and ESM-DBP[52] (which predicts DNA-binding proteins), represent another subset. These models excel at discovery and annotation tasks but mainly identify candidate scaffolds rather than optimize functional properties. Traditional ML-directed evolution (MLDE) approaches can be used to combine predictive models with manually designed libraries[31,32,34–37,53], making results sensitive to predictor accuracy, prior assumptions, and biases in the training data.

Generative models, by contrast, create novel sequences or structures tailored to specific objectives. Examples include structure-generating models such as RFdiffusion[54,55], which create protein backbones that scaffold functional motifs, and mutation proposing models such as FuncLib[56], which suggest active-site substitutions without manual library construction. However, such models have limitations: they often depend on high-quality structural information and abundant sequence homologs, and may fail for dynamic or allosterically regulated proteins. More recent approaches, such as preference-based learning (e.g., ProteinDPO[57]), incorporate relative comparisons between variants, but their scope is narrow—focusing on single objectives and lacking adaptability for tasks where tradeoffs between multiple properties must be considered. This is especially relevant for biosensors, where increasing sensitivity often reduces selectivity.

To overcome existing limitations, we set out to develop a directional, multi-objective ML model for engineering aTF-based biosensors that relies on a controlled extrapolation framework (Fig. 1)[58]. This model uses a sequence-to-sequence architecture to learn how amino acid sequence changes influence protein function, guided by tokens that encode the direction of property change. This approach eliminates the need for downstream predictors and allows direct manipulation of the model's latent space. Unlike preference-based models that only highlight improvements, our directional token approach exposes the model to both beneficial and detrimental mutations within the same training framework. For example, when the model sees a sequence pair labeled with 'decrease/increase' tokens, it simultaneously learns which mutations harm the first property while benefiting the second. This bidirectional learning provides a rich training signal about the mutational landscape[59].

We implement an active learning framework combining this directional ML model with a cell-free gene expression (CFE) system. The CFE system uses crude cellular extracts and reaction components (e.g., energy substrates, amino acids) to enable high-throughput transcription and translation of DNA templates outside living cells[60–62]. Building on our previous work[63,64], we integrate ML-guided design to achieve multi-objective optimization of the lead-responsive aTF PbrR, originally from the megaplasmid pMOL30 of *Cupriavidus metallidurans*[65]. Lead was selected because of its severe public health impact[66–68]. In the United States alone, there are an estimated 9.2 million lead service lines still in use[69]. Using the open, scalable CFE system, we rapidly generate positive and negative sequence-function data across multiple ligand conditions to train the model and iteratively refine predictions. This integrated, data-driven workflow enabled engineering of PbrR mutants that balance sensitivity and selectivity, meeting performance requirements for lead detection in drinking water.

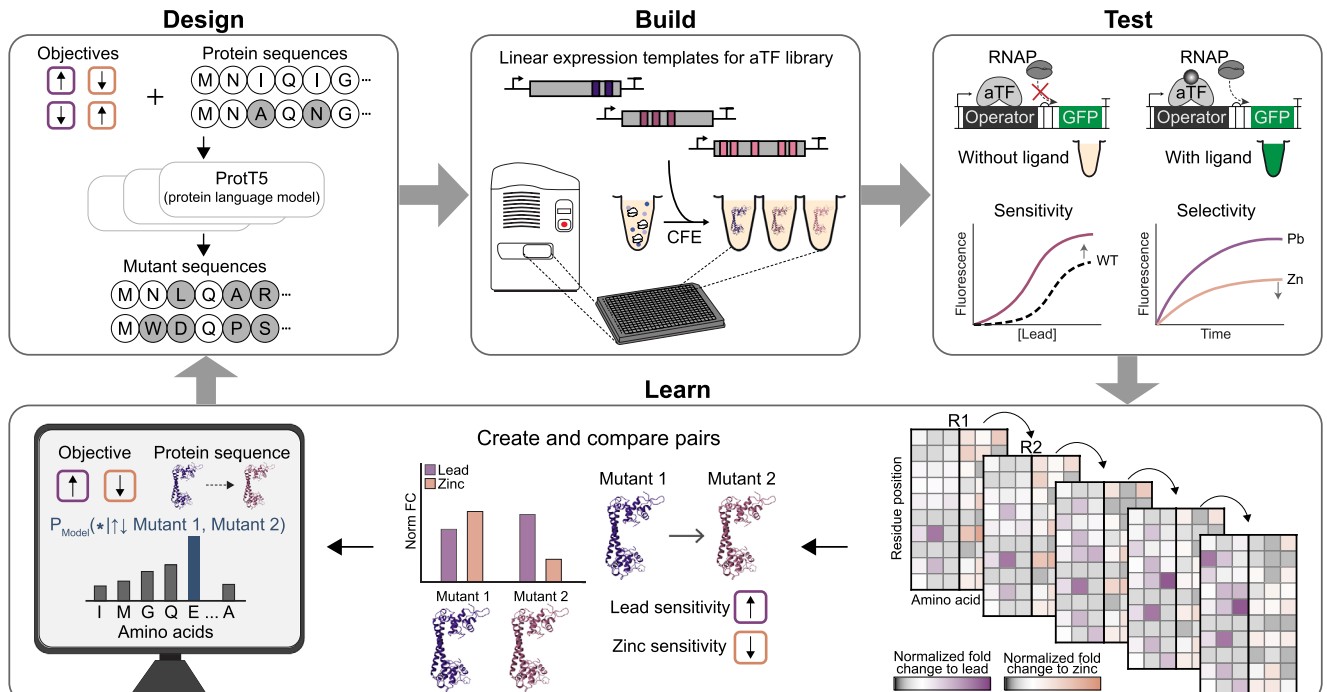

**Fig. 1 | An ML-guided, cell-free expression workflow for transcription factor-based biosensor development.** The schematic shows the Design-Build-Test-Learn workflow applied to rapidly tune the sensitivity and selectivity of aTFs, with PbrR as a model. The ML model is trained on paired mutant data comprised of sequence comparisons with directional objective labels to predict new mutant sequences (Design). Acoustic liquid handling robotics are used to set up CFE reactions for testing mutant libraries (Build). The libraries are screened for desirable lead biosensor characteristics (Test). Sequence-function data is used to create and compare pairs of mutants to identify amino acid residues with a high probability of being functionally important (Learn).

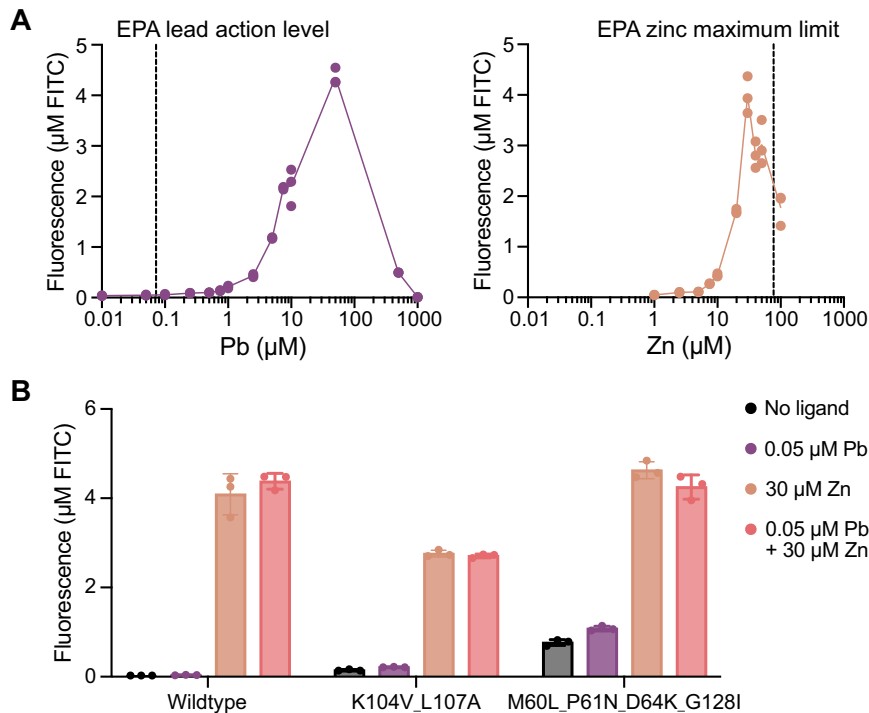

**Fig. 2 | A PbrR-based biosensor for lead detection in cell-free expression systems. A** Wildtype PbrR does not have the sensitivity to lead at the EPA action level (0.048 μM lead or 10 ppb)[71] and has high sensitivity towards levels of zinc below the EPA maximum limit (76 μM or 5 ppm)[73]. Data represent three biological replicates at each ligand concentration ($n = 3$). **B** We previously engineered PbrR mutants[64] to have sensitivity to lead at the EPA action level; however, these mutants have a strong response to zinc and give false positive results as a diagnostic for lead contamination in tap water. Data are presented as mean values +/− SD of three biological replicates ($n = 3$). Source data are provided as a Source Data file.

## Results

### PbrR as a model allosteric transcription factor for biosensor design

PbrR is a transcriptional activator that can function in a cell-free biosensing system to detect lead[64]. Upon ligand binding, PbrR undergoes a conformational change to distort the cognate operator site it is bound to, which initiates transcription of a downstream gene[70], such as the superfolder green fluorescent protein (sfGFP) reporter gene. However, wildtype PbrR does not have the sensitivity and selectivity requirements for diagnostic applications. The U.S. Environmental Protection Agency (EPA) action level for lead starting in 2027 will be 0.048 μM lead (10 ppb)[71] but the wildtype PbrR-based biosensor does not induce sfGFP expression in CFE systems until ~1 μM lead (Fig. 2A). Wildtype PbrR also reacts with other divalent ions, such as zinc[72]. Zinc is commonly found in tap water due to dissolution from pipes, and zinc concentrations below its EPA maximum limit (76 μM or 5 ppm)[73] strongly activate the wildtype PbrR-based biosensor (Fig. 2A). The cross-reactivity of PbrR towards lead and zinc has been studied. An in vivo directed evolution study improved lead selectivity of PbrR but did not address lead sensitivity. In addition, our previous engineering campaign improved lead sensitivity of PbrR to the EPA action level[64], but the best mutants also activate gene expression in the presence of zinc (Fig. 2B), limiting their utility for diagnostic applications due to false positives.

### Generating an initial dataset for ML model training

Towards diagnostic application requirements, we sought to simultaneously tune two biosensor characteristics using an ML-guided, design-build-test-learn (DBTL) workflow. This required an initial sequence-to-function dataset on biosensor activity towards lead and zinc to train the model (Fig. 3A).

To generate this data, we screened 1155 mutants that we previously created when engineering PbrR only for increased lead sensitivity[64]. These mutants were generated by an alanine scanning mutagenesis (145 mutants), site saturation mutagenesis (931 mutants), and selected combinational mutagenesis (79 mutants). Using liquid handling robotics to perform a plate-based high-throughput screen, we carried out 3465 unique reactions at 1 μL scale to test each mutant at a low concentration of lead (1 μM), high concentration of zinc (30 μM), and no ligand condition. The impact of each mutation from the alanine scanning mutagenesis (Fig. 3B) and site saturation mutagenesis (Fig. 3C) libraries on biosensor sensitivity relative to wildtype towards 1 μM lead and 30 μM zinc is represented in the heatmaps as normalized fold change (FC)[64]. For a mutation that was in a combinatorial mutant, the mean normalized FC of all mutants containing that mutation is represented. The mutability of PbrR for ligand sensitivity is not limited to the ligand binding domain, which highlights the difficulty of rational engineering of allosteric proteins. For example, mutations at residues in the DNA binding domain (e.g., M60, P61 and D64) and in the helix-turn-helix domain (e.g., K104 and L107) had diverse effects on activity.

A large shift in biosensor activity was not expected. Instead, mutations that result in small increases in lead sensitivity (lead normalized FC > 1) and small decreases in zinc sensitivity (zinc normalized FC < 1) would be important over iterative engineering rounds towards the goal of lead selectivity over zinc (Fig. 3D). An ideal mutant has a high normalized FC to lead and a normalized FC to zinc of zero. Because wildtype PbrR displays no activity towards 1 μM lead, a normalized FC to lead greater than one indicates increased sensitivity. In contrast, wild-type PbrR has strong activity towards zinc, so the normalized FC to zinc must be close to zero to indicate no zinc sensitivity. In this initial set, mutants generally displayed the same change in

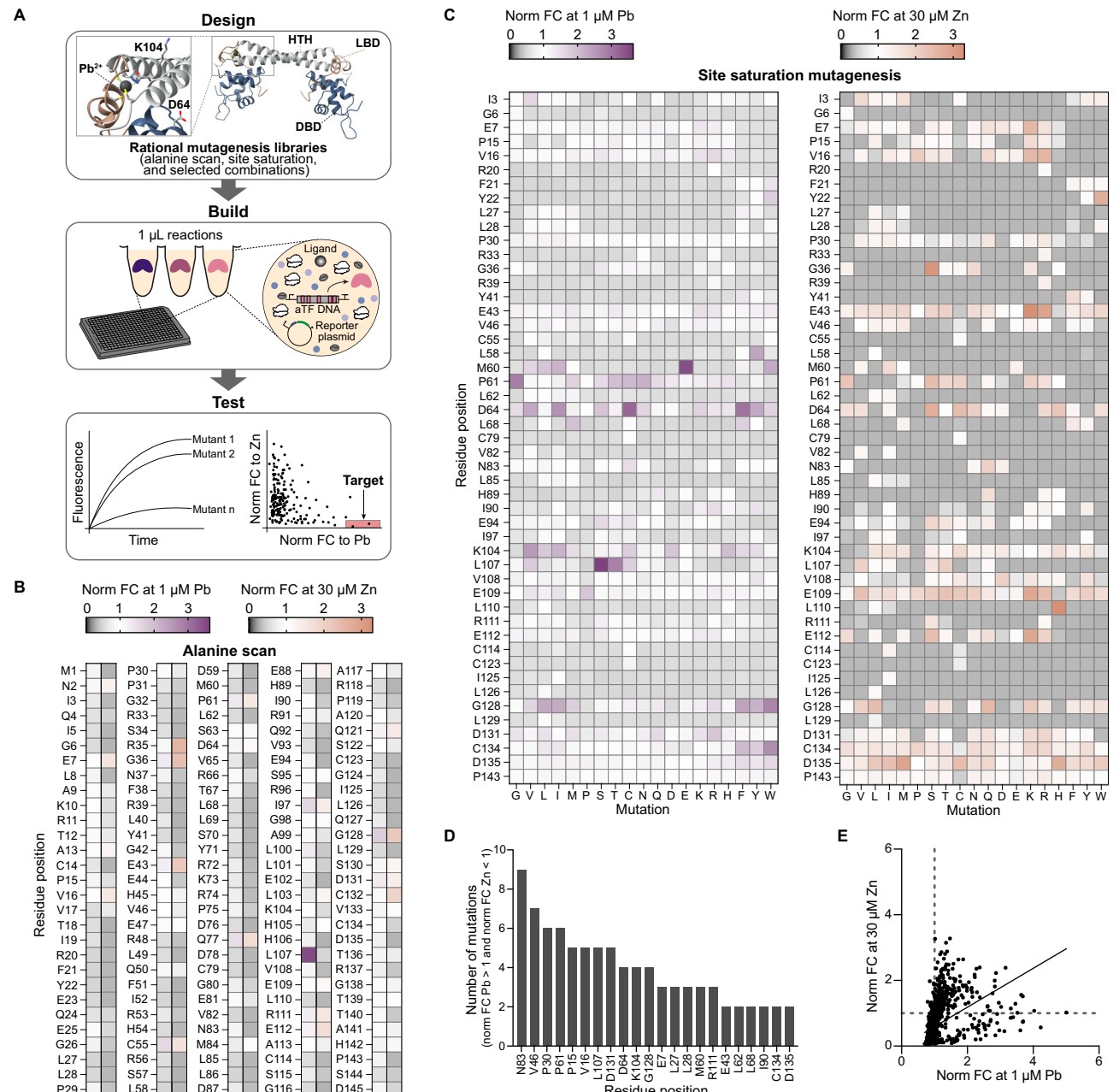

**Fig. 3 | Rapid generation of a sequence-function landscape dataset for ML-guided directed evolution of PbrR. A** Schematic of the cell-free workflow. Mutant libraries were rationally designed in a previous study[64] and are screened against low lead concentration (1 μM), high zinc concentration (30 μM), and no ligand condition using a high-throughput, plate-based assay. Mutant activity is measured as fluorescence and assessed by its fold change normalized to wildtype fold change ($n = 2$). The mean normalized fold change (norm FC) towards lead and zinc was calculated at every residue tested in the (**B**) alanine scanning mutagenesis library and (**C**) site saturation mutagenesis library. **D** Residue positions with multiple mutations towards lead selectivity over zinc are identified. **E** Scatterplot of individual mutant activity towards lead and zinc represented as the mean normalized FC of two biological replicates ($n = 2$). Source data are provided as a Source Data file.

activity towards lead and zinc, either sensitivity towards both ligands increased or decreased (Fig. 3E).

## Creating a paired dataset for the ML model
Our ML-model is trained on pairs of mutants, with each pair labeled according to the observed direction of functional change, rather than individual sequences that are used in traditional ML-methods. This is a departure from previous approaches and builds on the Iterative Controlled Extrapolation framework[58]. Pairing data also eliminates the need for numerical predictors, which can be unreliable when data is scarce. By training the model on sequence comparisons with directional labels, it learns generalizable patterns. We track how each functional label changes from one mutant to another, generating directional categories for each pair of mutants. For example, with two objectives, we consider four directional categories (e.g., increase/increase, increase/decrease, decrease/increase, and decrease/decrease), each assigned a unique token (Fig. 4A). As the number of objectives increases, the number of possible categories grows, but the model's task remains the same, which is to learn the sequence edits associated with each directional shift. This approach reduces the model's tendency to memorize specific sequences, improving its ability to extrapolate and suggest mutations in unsampled regions of the mutational landscape.

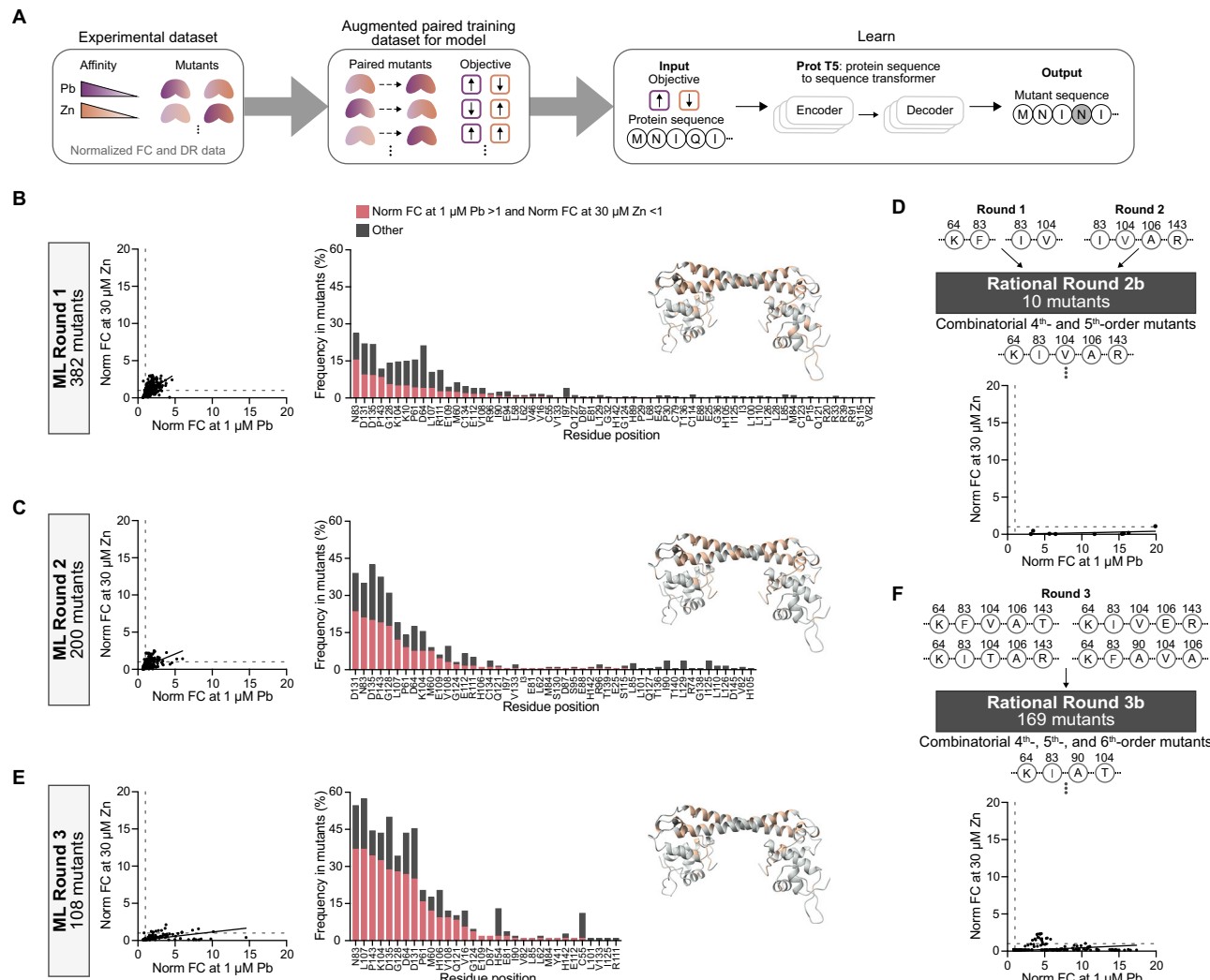

**Fig. 4 | ML-guided directed evolution of PbrR towards lead selectivity over zinc.** **A** For the learn step of our DBTL cycle, mutant data is paired, and each pair is labeled with an observed direction of functional change towards lead and zinc. A sequence-to-sequence large language model is trained on paired data to predict mutants. We performed five rounds of engineering with three ML-guided rounds (**B**, **C**, **E**) and two rational rounds using combinatorial mutagenesis strategies (**D**, **F**).

Over the rounds, we observe a large shift in mutants with increased normalized FC to lead and decreased normalized FC to zinc, indicating lead selectivity over zinc. As the model is trained on more data, it focuses on selected residue positions in its predictions. Mutant data in the screening assay were collected in biological triplicate ($n = 3$), and the mean normalized FC for each mutant is used in the scatterplots. Source data are provided as a Source Data file.

## Active learning-guided optimization of a cell-free biosensor based on PbrR

With an ML model framework designed and trained on an initial dataset (Round 0), the first round of PbrR engineering towards increased lead sensitivity and decreased zinc selectivity consisted of 382 computationally predicted mutants ranging from 1st- to 6th-order mutants (i.e., amino acid changes). This number was chosen to match the capacity of the 384-well plate that is used in our assay, while leaving wells open for controls. We screened these mutants against a low concentration of lead and a high concentration of zinc to identify mutants with lead selectivity over zinc. Like Round 0, we observed that most mutants displayed either complete loss of function or increased lead and zinc sensitivity (Fig. 4B). However, there were two mutants (D64K_N83F and N83I_K104V) that showed a higher fluorescent response to lead than zinc (Supplementary Fig. 1). We validated these mutants in experiments set up by hand to confirm the lead selectivity over zinc (Supplementary Fig. 2). The altered residues of these mutants aligned with the ones we previously identified in Round 0 as being important for lead selectivity over zinc, which motivated us to analyze

the residue exploration of the model. We calculated "Frequency in mutants" as the number of mutants that included a mutation at the specified amino acid residue. In Round 1, the model targeted 62 residues with a bias towards 11 residues that were each in at least 10% of mutants. In addition, we observed that mutating residues N83 and P143 more often resulted in increased lead sensitivity and decreased zinc sensitivity.

The two mutants from Round 1 that displayed lead selectivity over zinc had low fluorescent output, which is a limitation for use in a diagnostic. To address this, we began to train the model on normalized dynamic range (DR) in addition to normalized FC in Round 2. DR is the concentration of sfGFP synthesized in the absence of ligand (leak) subtracted from the concentration of sfGFP synthesized in the presence of ligand. We normalized the DR of each mutant to wildtype to more clearly observe relative activities and normalize for noise associated with assays run on different days. Negative normalized DR values were set to zero for model training. FC is a better sensitivity measure of mutants, while DR better reflects the signal response of mutants.

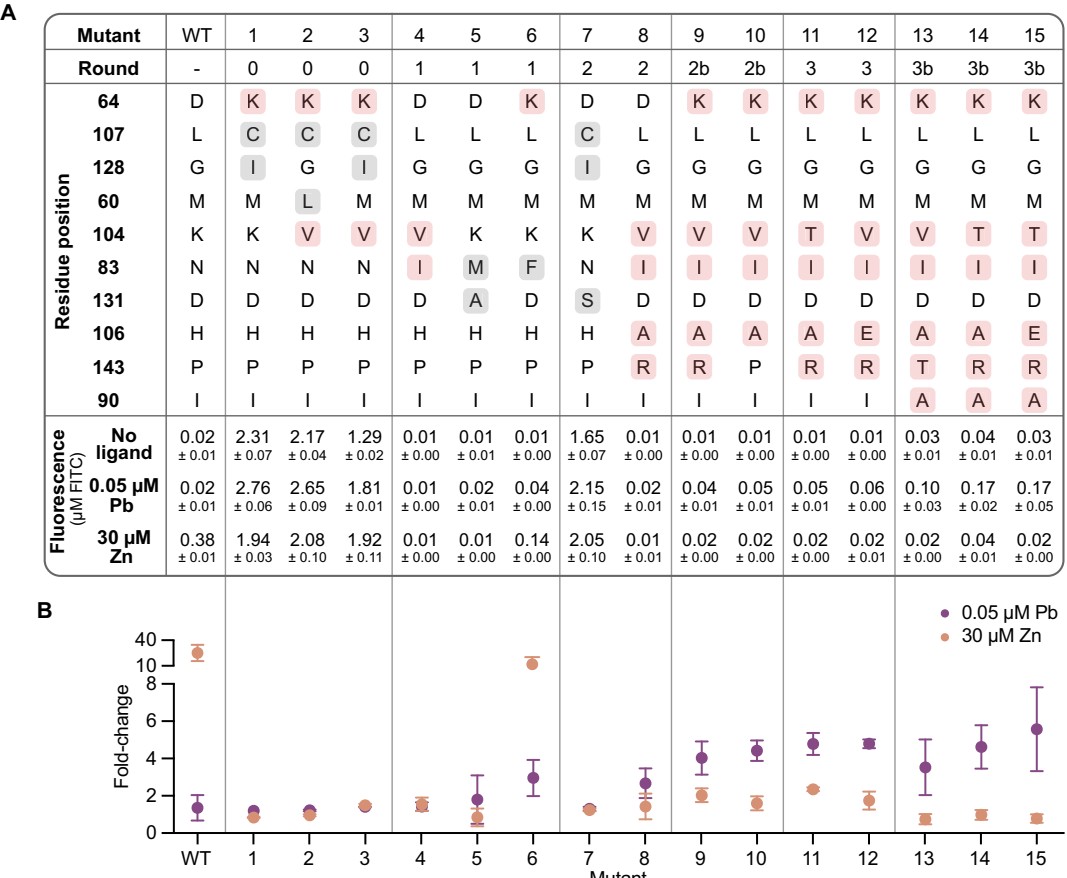

| Mutant | WT | 1 | 2 | 3 | 4 | 5 | 6 | 7 | 8 | 9 | 10 | 11 | 12 | 13 | 14 | 15 |
|---|---|---|---|---|---|---|---|---|---|---|---|---|---|---|---|---|
| Round | - | 0 | 0 | 0 | 1 | 1 | 1 | 2 | 2 | 2b | 2b | 3 | 3 | 3b | 3b | 3b |
| **Residue position** 64 | D | K | K | K | D | D | K | D | D | K | K | K | K | K | K | K |
| 107 | L | C | C | C | L | L | L | C | L | L | L | L | L | L | L | L |
| 128 | G | I | G | I | G | G | G | I | G | G | G | G | G | G | G | G |
| 60 | M | M | L | M | M | M | M | M | M | M | M | M | M | M | M | M |
| 104 | K | K | V | V | V | K | K | K | V | V | V | T | V | V | T | T |
| 83 | N | N | N | N | I | M | F | N | I | I | I | I | I | I | I | I |
| 131 | D | D | D | D | D | A | D | S | D | D | D | D | D | D | D | D |
| 106 | H | H | H | H | H | H | H | H | A | A | A | A | E | A | A | E |
| 143 | P | P | P | P | P | P | P | P | R | R | P | R | R | T | R | R |
| 90 | I | I | I | I | I | I | I | I | I | I | I | I | I | A | A | A |
| **Fluorescence (µM FITC)** No ligand | 0.02 ± 0.01 | 2.31 ± 0.07 | 2.17 ± 0.04 | 1.29 ± 0.02 | 0.01 ± 0.00 | 0.01 ± 0.01 | 0.01 ± 0.00 | 1.65 ± 0.07 | 0.01 ± 0.00 | 0.01 ± 0.00 | 0.01 ± 0.00 | 0.01 ± 0.00 | 0.01 ± 0.00 | 0.03 ± 0.01 | 0.04 ± 0.01 | 0.03 ± 0.01 |
| 0.05 µM Pb | 0.02 ± 0.01 | 2.76 ± 0.06 | 2.65 ± 0.09 | 1.81 ± 0.01 | 0.01 ± 0.00 | 0.02 ± 0.01 | 0.04 ± 0.00 | 2.15 ± 0.15 | 0.02 ± 0.01 | 0.04 ± 0.01 | 0.05 ± 0.01 | 0.05 ± 0.01 | 0.06 ± 0.00 | 0.10 ± 0.03 | 0.17 ± 0.02 | 0.17 ± 0.05 |
| 30 µM Zn | 0.38 ± 0.01 | 1.94 ± 0.03 | 2.08 ± 0.10 | 1.92 ± 0.11 | 0.01 ± 0.00 | 0.01 ± 0.00 | 0.14 ± 0.00 | 2.05 ± 0.10 | 0.01 ± 0.01 | 0.02 ± 0.00 | 0.02 ± 0.00 | 0.02 ± 0.00 | 0.02 ± 0.01 | 0.02 ± 0.00 | 0.04 ± 0.01 | 0.02 ± 0.00 |

**Fig. 5 | Mutations at six residues that accumulated in five rounds of PbrR engineering are important for lead selectivity over zinc. A** Table highlights the mutations found in the top mutants of each round. Mutations found in the final best mutants are highlighted in pink, while mutations only found in earlier rounds are highlighted in gray. The fluorescence and (**B**) fold change at 0.05 µM Pb and 30 µM Zn were measured in screen validation experiments. Data was collected in biological triplicate (*n* = 3). Mean fluorescence +/− SD is represented in (**A**). Mean fold change is plotted in (**B**) with error bars representing SD. Source data are provided as a Source Data file.

In Round 2, we screened 200 computationally predicted higher-order mutants, with 100 mutants predicted from the model trained on normalized DR data at lead and zinc and the other 100 mutants predicted from the model trained on normalized DR data at zinc and normalized FC data at lead. As the model became better informed by data from earlier rounds, fewer mutants were needed to explore the sequence space effectively. Testing 100 mutants per metric allowed us to compare the performance of each training strategy while reducing DNA synthesis costs.

Overall, we observed a modest increase in the number of mutants with a higher normalized FC to lead relative to zinc (Fig. 4C). However, most mutants still showed a stronger signal response to zinc than lead (Supplementary Fig. 1). Mutants predicted from the model trained on lead normalized FC and zinc normalized DR generally had higher leak. Despite this, the ML model began proposing hits with higher-order mutations that were not obvious or additive. For example, the model predicted the mutant N83I_K104V_H106A_P143R, which included the H106A mutation, a substitution not previously shown to increase lead sensitivity on its own. This mutant displayed a higher signal to lead than zinc and was validated in a by-hand experiment (Supplementary Fig. 2). In addition, we observed a decrease in the number of residues (48 residues, 33% of protein) explored and mutated in this round of computational predictions. The bias towards key residues, such as N83, highlights the importance in tuning activity towards increased sensitivity for lead and decreased sensitivity to zinc simultaneously.

We noted that the three "winners" from Round 1 and 2 contained overlap mutations at N83 and K104 and decided to perform a round of rational engineering, Round 2b, by creating ten higher order (4th- and 5th-order) mutants from the six unique mutations at five residue positions from previous winners, consisting of K64D, N83I or N83F, K104V, H106A, and P143R (Fig. 4D). These designs were informed directly by prior ML-guided rounds and experimental validation, reflecting a strategic recombination of high performing substitutions. While these rational mutants were not proposed de novo by the model, they drew directly from the higher order combinations that the model had already prioritized, which is a targeted exploitation of model discovered signals. Targeted exploitation of high-performing mutants through combination has been observed to improve sensor activity in our previous PbrR engineering efforts[64]. All mutants displayed significantly higher normalized FC to lead than to zinc. Importantly, the validation of these mutants showed that D64K_N83I_K104V_H106A and D64K_N83I_K104V_H106A_P143R exhibited lead selectivity over zinc at a lead concentration (0.05 µM) near the EPA action level of lead (0.048 µM) (Supplementary Fig. 2).

We next trained the ML model on Round 2b data and observed a significant improvement in the computational predictions because the training data now contained several mutants with better lead sensitivity than zinc sensitivity. For Round 3, the ML model was trained on two datasets: (i) normalized FC at lead and zinc, and (ii) normalized DR at lead and zinc. Overlap mutants that were predicted from both training datasets were tested. Of the 108 mutants screened, 75 mutants

had a normalized FC to lead greater than one and to zinc less than one (Fig. 4E) and 34 mutants had higher fluorescent signal to 1 µM lead than to 30 µM zinc (Supplementary Fig. 1). We were again able to screen a reduced library size of 108 mutants because the large dataset from previous rounds increased the reliability of model predictions. The ML model limited its exploration to 32 residues and strongly biased mutations at residues seen in Round 2b. The success of Round 2b motivated another rational round of combinatorial mutagenesis with the top 4 mutants from validating Round 3 hits (Supplementary Fig. 2). We designed a library of 169 higher-order mutants (4th-, 5th-, and 6th-order) at 6 residues with 10 unique mutations (Fig. 4F). In Round 3b, all mutants had a higher normalized FC to lead than to zinc and 137 mutants exhibited lead selectivity over zinc (Supplementary Fig. 1).

As our evolutionary scan traversed the fitness landscape, the diversity of mutated residues positions decreased. The top mutants from each round only covered 10 residues and 16 mutations in total (Fig. 5A), and the incremental additional of key mutations through the rounds increased mutant lead sensitivity at relevant lead concentrations while maintaining low zinc sensitivity (Fig. 5B). Although mutations L107C and G128I increase lead sensitivity, they also cause high leak and were disfavored in later rounds. Mutations D64K, K104V, and N83I were identified early in Rounds 0 and 1 as important for the desired biosensor characteristics. Then, mutations H106A and P143R shifted PbrR towards lead selectivity over zinc at a lead concentration 60-fold lower than the zinc concentration. Finally, the addition of mutation I90A in Round 5 eliminated zinc sensitivity. Single mutants H106A, P143R, and I90A did not show beneficial behavior, which highlights the ML model's ability to capture unusual combinations. Some of the mutations are also biophysically unexpected (e.g., D64K, P143R), further highlighting the power of the model.

The top mutants of Round 3b exhibited high sensitivity to lead at the EPA action level without activation by zinc, indicating low likelihood of false positive results to due zinc crosstalk. Importantly, the ML model and rational design served complementary roles throughout the engineering process. Rational design efficiently built on validated mutations by combining previously successful substitutions into higher-order mutants, while the model proposed unexpected non-additive combinations where some individual mutations alone offered little benefit. When rationally designed mutants were incorporated into subsequent ML training, they strengthened the model's confidence in key regions of the sequence space and improved its ability to prioritize synergistic mutations. This exchange between computationally proposed and experimentally guided recombination enabled more focused searches and was effective in realizing the final design.

### Freeze-dried, cell-free PbrR biosensors as a diagnostic assay for lead in drinking water

Cell-free biosensors are promising point-of-use diagnostics for water contamination because they can be lyophilized for stable storage and transportation and then rehydrated with the water sample[9,74]. We next tested if our PbrR-based biosensor could be freeze-dried and remain functional to detect lead in water samples (Fig. 6A). Instead of aTF expression from a DNA template during a CFE reaction, we created extracts enriched for the aTF (i.e., the aTF was expressed in the extract source strain prior to cell lysis) to improve biosensor performance (e.g., increased sensitivity and dynamic range)[8,64]. We made enriched extracts with the top three mutants from Round 3b (Supplementary Fig. 3), and mutant D64K_N83I_I90A_K104T_H106A_P143R (Fig. 6B) exhibited the best biosensor performance with low leak and the highest signal response to 0.05 µM lead (Fig. 6C).

To evaluate the potential of this mutant as a diagnostic, we next demonstrated that lyophilization has no negative impact on sensor function (Fig. 6D), and that the mutant only exhibits cross-reactivity to mercury when tested against a panel of divalent metal ions that may be found in municipal water (Supplementary Fig. 4). Then, based on previous work for developing point-of-use water quality diagnostics[75], we modified the reporter system of the biosensor to express the enzyme catechol 2,3-dioxygenase (C23DO), which cleaves colorless catechol into the yellow pigment 2-hydroxymuconate semialdehyde (Fig. 6E). Using an enzymatic, colorimetric reporter improves the kinetics of the reaction[64] and provides a visible difference between ligand conditions (Fig. 6F). We rehydrated lyophilized sensor reactions with municipal water samples that were collected in Evanston and Chicago, Illinois. Metal concentrations in these samples were quantified using Inductively Coupled Plasma Mass Spectrometry (ICP-MS). We observed both quantitative and qualitative differences in sensor responses between the lead-free water sample and those containing lead (0.03 – 0.13 µM; 5.71 – 26.41 ppb), indicating that this PbrR-based biosensor could be used as an effective point-of-use diagnostic for detecting lead at the legal limit in real-world water samples (Fig. 6G).

## Discussion

In this work, we established a directional, multi-objective ML-guided cell-free platform for tuning multiple transcription factor biosensor characteristics simultaneously. The ML model uniquely adapted a controlled extrapolation framework for multi-parameter optimization and trained a sequence-to-sequence language model on paired mutant data. Using a high-throughput CFE screening assay, we showcased the ability to rapidly screen libraries to engineer lead-responsive transcription factor PbrR to have lead selectivity over zinc at lead concentrations at the EPA action level. Being able to discriminate between two similar divalent cations, $Pb^{2+}$ and $Zn^{2+}$, is an important example of using ML-guided methods for protein design.

A key feature of our work is the high efficiency of the ML model to optimize over multiple objectives, as it enabled us to screen less than 1% of the search space to identify mutants with the desired selectivity and high sensitivity to lead. Our ML framework is especially suitable for design problems where optimizing one function may negatively affect another, requiring explicit definition of multiple objectives. It is also advantageous in data-scarce situations, as the use of paired data synthetically expands the training set. In addition, by eliminating the need to tune or retrain separate predictors for each new objective, our method provides flexible multi-objective optimization within a single model. By incorporating directional tokens, the model can be flexibly prompted to pursue any design objective over any scale, generating any desired number of candidates tailored to experimental capacity. This enables efficient exploration of vast mutational landscapes without manual intervention.

Through efforts to train our ML model and then subsequent screening rounds, we assayed 2024 mutants that explored all residue positions to gain an understanding of the positive and negative sequence-function landscape of PbrR. By thoroughly optimizing the high-throughput workflow[63], we generated high-quality data to train a model to recognize patterns that would be difficult to do without computational tools. For example, mutations at residue I90 provided little to no shift in activity towards lead selectivity over zinc in Rounds 1 and 2. In workflows that rely only on rational methods, such as combinatorial mutagenesis, this residue would likely not have been explored in later rounds but proved essential for our goal in Round 5.

At the end of the engineering campaign, we identified 6 unique residue positions that need to be mutated together to shift the selectivity of PbrR towards lead and away from zinc. These residues span all domains of PbrR: D64 is in the DNA-binding domain (DBD), N83, I90, K104, and H106 are in the helix-turn-helix domain (HTH), and P143 is in the ligand-binding domain (LBD). The distribution of the mutations highlights the limitations of using rational engineering approaches for allosteric proteins, as it is difficult rationalize why this specific combination of mutations would be important for tuning ligand selectivity. We hypothesize that these mutations are impacting metal ion coordination, DNA affinity, homodimerization, and allostery. For example,

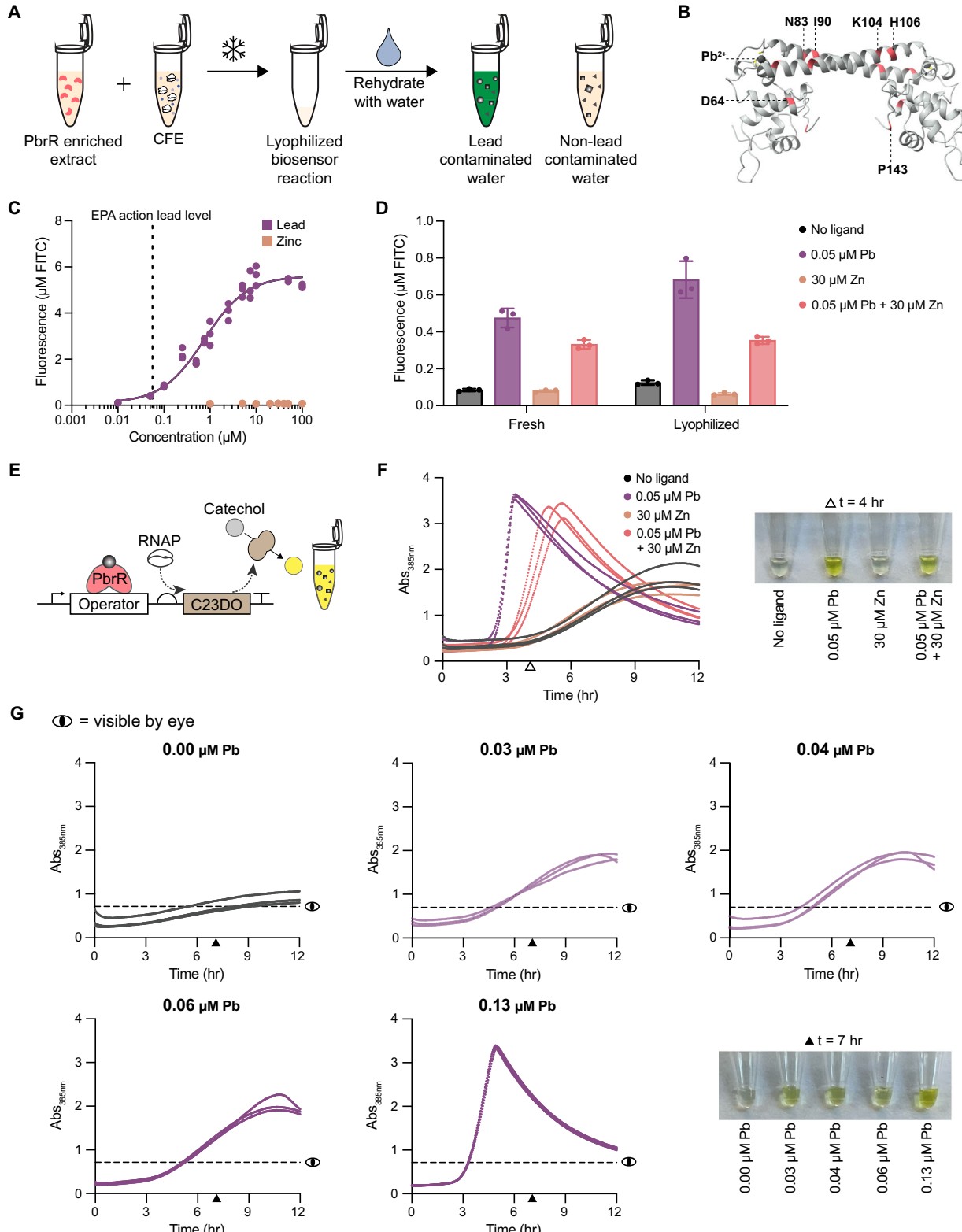

**Fig. 6 | Freeze-dried cell-free biosensor based on mutant PbrR detects lead without zinc activation. A** Schematic of the PbrR-based biosensor as a point-of-use diagnostic. Cell-free biosensing reaction supplemented with an extract enriched with PbrR is lyophilized. Lyophilized reactions are rehydrated with water samples. **B** Residues on PbrR structure are highlighted to demonstrate the distribution of mutations across the protein for the best mutant D64K_N83I_I90A_K104T_H106A_P143R. **C** Dose response curves for the best performing biosensor mutant show high selectivity for lead over zinc at relevant lead concentrations. Hill slope for lead dose-response is 0.94. **D** Lyophilization does not negatively impact biosensor function. Data are presented as mean values +/− SD of three biological replicates ($n = 3$). **E** Schematic of PbrR-based biosensor using the enzymatic, colorimetric catechol reporter. **F** PbrR-based biosensor using catechol reporter functions after lyophilization and rehydration. A visible difference is seen in ligand conditions at $t = 4$ hrs. **G** Lyophilized biosensor reactions with the catechol reporter are rehydrated with municipal water samples containing a range of lead concentrations. All data shown were collected in biological triplicate ($n = 3$). Supplementary Fig. 5 and Supplementary Fig. 6 detail how the 'visible by eye' line is determined. The time point at 7 h was selected to discriminate between samples with lead and the no-lead control. Source data are provided as a Source Data file.

$Pb^{2+}$ typically has more flexible coordinate geometries with proteins compared to $Zn^{2+}$, which prefers tetrahedral coordination[76,77]. The mutations we identified may subtly reshape the binding pocket's geometry or electrostatics to disfavor zinc coordination and improve lead coordination, as well as increase cross reactivity to mercury (Supplementary Fig. 4). In addition, mutations in the HTH motif may influence the allosteric communication between the LBD and DBD domains, altering the transcriptional response of the biosensor to different metals[78]. More broadly, rapidly building datasets to navigate vast protein sequence space remains difficult. We expect that our dataset will help support general advances in ML-model development for synthetic biology.

We anticipate that our approach to tune biosensor characteristics can be applied to any transcription factor but may require modifications to the experimental set-up and ML model parameters. For example, when working with a repressor transcription factor, an additional incubation step when setting up a CFE biosensing reaction may be required to allow for the transcription factor to bind to its operator site to reduce high leak. Depending on the mutability of a transcription factor and the degree of targeted change in biosensor activity, various ML parameters will need to be explored. A recent study showed that thermodynamic and kinetic models were able to provide mechanistic reasoning for allosteric modulation of ligand selectivity in transcription factor MAX. Binding mechanisms were revealed in kinetic measurements[79]. Incorporating these types of models and data collection into our workflow could be beneficial in engineering transcription factor biosensors.

In terms of applications, just as freeze-dried CFE systems can be used for manufacturing[80-82] and education[83-86], our engineered PbrR-based biosensor holds promise for point-of-use diagnostics. For example, the system can be freeze-dried for easy storage, distribution, and activation by just adding water. In addition, the system is low-cost (i.e., ~10 cents per 15 µL reaction[80]). Furthermore, we showed a 500-fold improvement in sensitivity from a previous report[9], while avoiding Zn selectivity problems that have plagued past efforts with false positives[64]. Finally, we demonstrated that the biosensor works in real-world municipal water samples to detect lead. Future improvements will seek to accelerate time to response to minutes instead of hours, as has been accomplished with the ROSALIND system[9].

In sum, our ML-guided, cell-free workflow improved the process of exploring sequence-function search space to tune transcription factor-based biosensor characteristics by overcoming traditional directed evolution challenges with allosteric proteins and limitations of predictive scoring ML models. Looking forward, we anticipate that our active learning approach will accelerate the development of specialized diagnostics, and perhaps any engineered protein, for numerous synthetic biology applications.

## Methods

### Multi-objective controlled extrapolation

We extended the Iterative Controlled Extrapolation (ICE) framework[58] to enable multi-objective protein design without relying on numerical downstream predictors. ICE is a transformer based language model that performs single-objective, iterative rounds of sequence refinement. In each round, the model predicts small edits to a sequence to gradually achieve attribute values beyond the training distribution. Our approach leverages a sequence-to-sequence transformer trained directly on paired mutant comparisons, where each pair is labeled with a discrete token indicating the direction of functional change across one or more objectives. Unlike traditional iterative design strategies that involve multiple or iterative rounds of generation, our approach uses only a single generation step. This generation is seeded from a diverse set of experimentally validated sequences that meet specific starting criteria (e.g., high lead response and low zinc response). By avoiding multiple rounds, we reduce the risk of the model drifting away from experimentally reliable regions of sequence space. This encourages novelty through diversity in the initial seeds and enables controlled extrapolation to new designs while maintaining proximity to known data. A pseudocode representation of the model training and sequence generation procedures is provided in the Supplementary Information as Supplementary Note 1 and Supplementary Note 2, respectively. The model architecture, along with the training and generation workflows, is illustrated in Supplementary Fig. 7. These algorithms are described in detail in the following sections.

### Training data construction

Let $X \in A^L$ denote a protein sequence of length $L$, where $A$ is the set of all amino acid single-letter codes. Let $f$ be the unknown multi-objective function that maps each sequence to a vector of $j$ scalar measurements, where $j$ is the number of objectives. In our case, we define:

$$f(X) = \left[ f_{Zn}(X), f_{Pb}(X) \right] \tag{1}$$

where $f_{Zn}(X)$ and $f_{Pb}(X)$ denote the measured fold change or response to zinc and lead, respectively. Note that the model is trained without access to these numerical values. They are used only to assign directional labels. We define the training data as a set of $N$ sequence pairs $\left\{ \left( X_1^{(i)}, X_2^{(i)} \right) \right\}_{i=1}^{N}$, where each pair represents edits in sequence or mutations from $X_1$ to $X_2$, which may involve one or more amino acid substitutions. From a set of $M$ experimental samples, up to $\binom{M}{2}$ unique pairs can be formed, enabling substantial data augmentation, especially when $M$ is small. To ensure biological relevance and robust training, we include only those sequence pairs where the measured change in objective exceeds the estimated experimental noise threshold $\tau_\kappa$. For each pair $(X_1, X_2)$ and each objective $\kappa$, we assign a direction label:

$$label_\kappa = \begin{cases} \langle inc \rangle, f_\kappa(X_2 - X_1) > \tau_\kappa \\ \langle dec \rangle, f_\kappa(X_2 - X_1) \leq \tau_\kappa \end{cases} \tag{2}$$

Each label vector $d^{(i)} \in \{ \langle inc \rangle, \langle dec \rangle \}^j$ is prepended to the input sequence as a set of tokens. We ensure approximate balance across all $2^j$ possible combinations of directional labels to avoid training bias.

### Model architecture and training

We use a transformer encoder-decoder model based on the T5 architecture, specifically the ProtT5-XL-UniRef50 model from Rostlab[87]. We added two special tokens, $\langle inc \rangle$ and $\langle dec \rangle$, to represent directionality for each objective.

During training, the model receives an input prompt consisting of the directional tokens followed by the amino acid sequence $X_1$, and it is trained to autoregressively generate the output sequence $X_2$. The conditional probability of generating $X_2$ given the input is modeled as:

$$P_\theta \left( X_2 |, d, X_1 \right) = \prod_{t=1}^{L} P_\theta \left( X_{2,t} | d, X_1, X_{2, <t} \right), \tag{3}$$

where $X_{2,t}$ is the amino acid at position $t$ in the target sequence $X_2$, $X_{2, <t}$ is the partial sequence of $X_2$ up to but not including position t, $\theta$ is the model parameters learned during training, and L is the length of the target sequence. The model is trained to maximize the conditional log-likelihood of the target sequences across the dataset of N labeled sequence pairs:

$$L(\theta) = \sum_{i=1}^{N} \log P_\theta \left( X_2^{(i)} | d^{(i)}, X_1^{(i)} \right) \tag{4}$$

where $X_1^{(i)}$, $X_2^{(i)}$, $d^{(i)}$ are the source sequence, target sequence, and directional label for the $i$-th training example, respectively. Loss is computed via token-level cross-entropy, and the model is optimized using AdamW with a learning rate of 1e-4 and weight decay of 1e-4, using Hugging Face's Seq2SeqTrainer. During training, to assess whether the model is generating valid sequences consistent with the training distribution, we use the SacreBLEU score[88].

To minimize memory usage, we apply the Low-Rank Adaptation (LoRA) framework for parameter-efficient finetuning[89]. Low-Rank Adaptation (LoRA) is used to efficiently finetune large pretrained language models without updating all weights. By injecting trainable low-rank matrices into frozen transformer layers, LoRA drastically reduces the number of parameters that need to be learned, saving memory and computation while allowing the model to adapt to new tasks or datasets. We set the rank of the update matrices to 16, applied a LoRA scaling factor of 32, and used a dropout probability of 0.05 in the LoRA layers. More details on model training and parameters are shown in Supplementary Table 1.

### Inference and design
At inference time, a set of seed sequences are provided along with the desired directional token (e.g., $\langle inc \rangle$ $\langle dec \rangle$ for increasing lead response while decreasing zinc response). For our first computational round, we use as seeds all sequences that show higher lead response and lower zinc response than the wildtype. The model then generates new sequences conditioned on the prompt.

Because the model operates directly in sequence space, it can be flexibly prompted with any combination of direction tokens without retraining. We use top-k sampling ($k = 10$) to generate diverse candidates. For each seed sequence, 20 candidates were proposed, and we filter by edit distance or number of mutations from the wildtype sequence as needed. This framework enables controllable and scalable exploration of the mutational landscape, guiding design toward functional improvement under user-define multi-objective constraints.

### DNA library generation
The DNA for wildtype PbrR was from Addgene (ID 167215). The protein sequence for this wildtype PbrR was from *Cupriavidus metallidurans* (Uniprot Q58AJ5). In this plasmid, the T7 promoter is used to drive expression of the PbrR gene. All aTF DNA sequences used in this study follow this design with changes only to the PbrR sequence to create the mutants. All DNA sequences can be found in the Supplementary Data 1 file. Plasmid DNA from Twist Biosciences was purchased for the alanine scanning mutagenesis library. For all other mutant libraries, eBlocks from Integrated DNA Technologies (IDT) were ordered with homology to the pJL1 backbone. The pJL1 backbone was ordered as a gBlock from IDT and amplified via PCR. The eBlocks and pJL1 backbone were assembled into plasmids using standard Gibson Assembly methods with a 30 min incubation at 50 °C.

The cell-free generation of mutant libraries were prepared based on a previously described method[63]. Briefly, with the commercially purchased or Gibson assembled plasmids as the DNA template, linear expression templates (LET) were generated via PCR reaction using Q5 Hot Start DNA Polymerase (NEB) in 384-well PCR plates (Bio-Rad). The primers used to generate the LETs were 5' CGA-TAAGTCGTGTCTTACCG 3' and 5' GCATAAGCTTTTGCCATTCTC 3'. LET yields were quantified using QuantiFluor dsDNA System (Promega). The Echo 525 was used to normalize LET DNA to a concentration of 4.5 ng/μL (5 nM). All transfer steps between plates, except for the Echo normalization step, were done using an Integra VIAFLO liquid handling robot.

### Cell extract preparation
Extract from BL21 Star™ (DE3) (Thermo Fisher Scientific C601003) optimized for endogenous transcription machinery was prepared based on previous reports[90–92]. The reporter plasmids used in this study are regulated under bacterial σ[70] promoters, and for cell-free gene expression of these plasmids, the extracts were processed with ribosomal runoff reaction and subsequent dialysis. In summary, an overnight culture was used to inoculate 2xYTP media (16 g/L tryptone, 10 g/L yeast extract, 5 g/L sodium chloride, 7 g/L potassium phosphate dibasic, 3 g/L potassium phosphate monobasic, pH 7.2) to a target starting optical density at 600 nm ($OD_{600}$) of 0.05. The culture was grown at 37 °C, shaking at 250 rpm. At $OD_{600} = 0.5$, isopropyl ß-D-1-thiogalactopyranoside (IPTG) at a final concentration of 1 mM was used to induce expression of T7 RNA polymerase. The cells were grown to an $OD_{600}$ of 3.0 before being harvested and centrifuged at $5000 \times g$ for 15 min at 4 °C. The resulting cell pellet was washed three times with 25 mL of cold wash buffer (14 mM magnesium glutamate, 60 mM potassium glutamate, 10 mM of Tris base, pH 7.8). Cells were pelleted between each wash step via centrifugation at $10,000 \times g$ for 2 min. After pouring off the supernatant of the final wash step, the cell pellet was weighed and resuspended in 1 mL of wash buffer per gram of cell pellet. Cells were then lysed with a single pass through an Avestin EmulsiFlex-B15 homogenizer at 20,000–25,000 psig. The lysed sample was centrifuged for 10 min at $12,000 \times g$ at 4 °C. The resulting supernatant underwent runoff by wrapping the tubes in aluminum foil and incubating them at 37 °C with shaking at 250 rpm for 1 hr. The sample was centrifuged again for 10 min at $12,000 \times g$ at 4 °C and the resulting supernatant is dialyzed for 3 hr 4 °C using a 10 K MWCO dialysis membrane, slowing spinning in dialysis buffer (14 mM magnesium glutamate, 60 mM potassium glutamate, 5 mM Tris base, 1 mM DTT, pH 8.0). After dialysis, the sample was centrifuged for 10 min at $12,000 \times g$ at 4 °C, and the supernatant (cell extract) was aliquoted, flash frozen, and stored at −80 °C.

For extracts enriched with a PbrR mutant, the above method was followed with some modifications. Within the same week as extract preparation, BL21 Star™ (DE3) cells were transformed with a sequence-verified plasmid of the mutant and plated on LB agar plates containing 50 mg/mL Kanamycin. Overnight cultures were grown with 50 mg/mL Kanamycin. During cell growth in 2xYTP media (no antibiotic), cells were induced at $OD_{600}$ of 0.5 with 0.5 mM IPTG to induce PbrR mutant expression and grown for 2 h post-induction. After washing the cells three times and resuspending in wash buffer, cells were lysed using the QSonica Q125 sonicator with a 3.175 mm diameter probe at a frequency of 20 kHz and 50% amplitude by 10 s ON/OFF pulses for two rounds of 60 s (delivering ∼ 400 J per round). The samples were kept on ice for 10 min between sonication rounds. After sonication, the lysed cells were centrifuged for 10 min at $12,000 \times g$ at 4 °C. The resulting supernatant was processed via runoff and dialysis and then flash frozen for storage at −80 °C.

### Cell-free expression biosensing reactions
Similar to previous works[64,90], CFE reactions were carried out using the PANOx-SP system[93–95]. To inhibit nuclease activity, 30 μg/mL GamS (NEB) was added to reactions with LETs. The reporter plasmids (pPbrR-sfGFP, Addgene ID 167222 and pPbrR-XylE (C23DO enzyme), Addgene ID 167254) were purified from overnight cultures using Qiagen HiSpeed® Plasmid Maxi Kit. Echo-assisted assembly of 1-μL CFE reactions were performed[64], and after 15 hr incubation at 30 °C, sfGFP was quantified by measuring fluorescence on a Biotek Synergy Neo2 plate reader at excitation of 485 nm and emission of 528 nm. By-hand validation experiments were carried out using aTF LETs purified with the Zymo DNA Clean & Concentrator kit as 10-μL reactions in black 384-well plates, clear flat-bottom plates (Greiner #781906). These reactions were incubated for 15 hr at 30 °C in a BioTek Synergy H1 plate reader with reads every 5 min at excitation of 485 nm and emission of 528 nm. Bar charts represent endpoint data at 15 hr. For each enriched extract, volume (% v/v) optimizations were performed. Reactions using enriched extracts were also done as 10 μL reactions in 384-well plates

(Greiner #781906) and incubated in a BioTek Synergy H1 plate reader for kinetic data collection. Fluorescence was quantified by fluorescein isothiocyanate (FITC) standard curves (Sigma-Aldrich 46950), which were created via dilutions in 50 mM sodium borate at pH 8.5. To quantify the color changes in sensing reactions using the catechol reporter, 10 μL reactions in 384-well plates (VWR 76446-984) were incubated in a BioTek Synergy H1 plate reader at 30 °C for 12 h with reads every 2 min at absorbance 385 nm for kinetic data collection. Pictures of the catechol reactions were taken using an iPhone 13 Pro. Lead solutions were prepared from lead chloride powder (Sigma-Aldrich 268690), and zinc solutions were prepared from zinc acetate powder (Sigma-Aldrich 383317).

## Lyophilization and rehydration of cell-free biosensing reactions

CFE reactions for lyophilization were set up as described above and lyophilization was performed as reported in literature[80]. Briefly, 35 μL or 15 μL reactions were aliquoted into 0.2 mL PCR tubes (Thermo Scientific AB-2000) with a hole punctured in the cap by an 18-guage needle. Samples were flash-frozen in liquid nitrogen and quickly transferred to the manifold adapter on a VirTis AdVantage Pro Freeze Dryer. Lyophilization was performed at 100 mTorr with the condenser set to − 80 °C. After lyophilization for 16–20 h, samples were rehydrated with water and pipette-mixed.

## Municipal water collection and analysis

For this work, the 1st and 5th liter municipal water samples were collected in 1 L wide-mouth HDPE bottles from households in Evanston and Chicago, IL. Non-acidified samples were used to test PbrR biosensing systems. Within two weeks of sample collection, 15 mL aliquots of each water sample were acidified to pH < 2 using ultra-pure $HNO_3$ (Ultrex, J.T. Baker, 67–70%). Samples were then analyzed via ICP-MS within 16 h after acidification, per EPA Method 200.8[96]. ICP-MS data were generated using a ThermoFisher iCAP-Q in kinetic energy discrimination mode with helium as the collision gas. The instrument was equipped with a CETAC ASX260 autosampler, calibrated with ten standard solutions and an acid blank. Standard solutions were made from two separate multielement stock solutions in 2% $HNO_3$: (i) major municipal water cations with concentrations from 0 to 10 mg/L like calcium and magnesium and (ii) trace elements with concentrations from 0 to 10 μg/L, like lead. The average limit of detection (LOD) for $^{206\text{-}208}Pb$ was 0.59 ng/L. Samples with lead concentrations initially measured above the calibration range were re-measured after dilution by the autosampler. The certification of the results were based on analyzing the standard reference material Aqua-1[97].

## Data collection and analysis

Data in this manuscript represent $n = 3$ biological replicates unless otherwise noted in the text or figure legends. All data were collected using stated instruments and associated commercially available software. Commercial software used includes: Gen5 Version 2.09.2 (BioTek Synergy Neo2 or H1) for measuring GFP fluorescence or absorbance and Qtegra ISDS (ThermoFisher iCAP-Q) for measuring municipal water content. Data analysis and figure generation were conducted using Excel Version 16.19.1, ChimeraX Version 1.9[98], Prism Version 10.4.2, Jupiter Notebook Version 7.2.2, and Chai-1[99].

## Statistics and reproducibility

No statistical method was used to predetermine sample size. No data were excluded from the analyses. The experiments were not randomized. The Investigators were not blinded to allocation during experiments and outcome assessment.

## Reporting summary

Further information on research design is available in the Nature Portfolio Reporting Summary linked to this article.

## Data availability

DNA sequences for the PbrR mutants used in this study are included in the Supplementary Data 1 file. The Uniprot accession code for wildtype PbrR is Q58AJ5. The Addgene accession code for the wildtype PbrR plasmid is 167215. The Addgene accession codes for the sfGFP and catechol reporter plasmids are 167222 and 167254, respectively. Source data are provided in this paper.

## Code availability

The code used in this manuscript is available at: Main code[100]: https://github.com/ShuklaGroup/multiobjective_controlled_extrapolation
Data used in code demo: https://uofi.box.com/s/qpaatf9ge9f3ofqq7aybqwkxkid94fgd

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

## Acknowledgements

This work was supported by the Army Research Laboratory and the Army Research Office (W911NF-23-1-0334 and W911NF-22-2-0246 to M.C.J.), AFOSR (FA9550-23-1-0420 to M.C.J.), and National Science Foundation (2319427 and 2310382 to J.B.L.). B.M.W. was supported by a National Science Graduate Research Fellowship (DGE-2234667). D.S. acknowledges support from NIH grant R35GM142745. Any opinions, findings, conclusions or recommendations expressed in this material are those of the authors and do not necessarily reflect the views of the National Science Foundation or the Department of Defense. Metal analysis was performed at the Northwestern University Quantitative Bio-element Imaging Center, generously supported by NASA Ames Research Center NNA06CB93G.

## Author contributions

B.M.W., N.C., D.S., and M.C.J. conceived this project. B.M.W., N.C., and H.M.E. planned experiments, prepared reagents, performed experiments, developed models, and/or analyzed the data. D.M.B., T.J.L., S.F., and V.B. collected municipal water samples. G.D. and J.-F.G. analyzed municipal water samples. J.B.L., A.S.K., D.S., and M.C.J. supervised the research. B.M.W., N.C., J.B.L., A.S.K., D.S., and M.C.J. contributed to the writing of the manuscript.

## Competing interests

The authors declare the following competing interest(s): M.C.J. is a co-founder and has interest in Stemloop, Inc., Pearl Bio, Gauntlet Bio, and Synolo Therapeutics; J.B.L. is a co-founder and has interest in Stemloop. Inc. These interests are reviewed and managed by Northwestern University and Stanford University in accordance with their conflict-of-interest policies. All other authors report no competing interests.

## Additional information

[1]Department of Bioengineering, Stanford University, Stanford, CA, USA. [2]Department of Chemical and Biomolecular Engineering, University of Illinois at Urbana-Champaign, Urbana, IL, USA. [3]Department of Chemical and Biological Engineering, Northwestern University, Evanston, IL, USA. [4]Chemistry of Life Processes Institute, Northwestern University, Evanston, IL, USA. [5]Center for Synthetic Biology, Northwestern University, Evanston, IL, USA. [6]Department of Civil and Environmental Engineering, Northwestern University, Evanston, IL, USA. [7]Department of Biomedical Engineering, Northwestern University, Evanston, IL, USA. [8]Bridges/Puentes: Justice Collective of the Southeast, Chicago, IL, USA. [9]Center for Water Research, Northwestern University, Evanston, IL, USA. [10]Interdisciplinary Biological Sciences Graduate Program, Northwestern University, Evanston, IL, USA. [11]Department of Bioengineering, University of Illinois at Urbana-Champaign, Urbana, IL, USA. [12]Department of Chemistry, University of Illinois at Urbana-Champaign, Urbana, IL, USA. [13]Beckman Institute for Advanced Science and Technology, University of Illinois at Urbana-Champaign, Urbana, IL, USA. [14]These authors contributed equally: Brenda M. Wang, Nicole Chiang. ✉e-mail: mjewett@stanford.edu

