## [Peer Review File · Nature Communications]

Active learning-guided optimization of cell-free biosensors for lead testing in drinking water

Corresponding Author: Professor Michael Jewett

Version 1:

Reviewer comments:

Reviewer #2

(Remarks to the Author)

This manuscript presents a compelling integration of machine learning (ML) and cell-free gene expression (CFE) to engineer allosteric transcription factors (aTFs) for improved biosensor performance. The work addresses a critical challenge in synthetic biology: tuning multiple biosensor parameters (e.g., sensitivity and selectivity) simultaneously. The authors demonstrate the utility of their approach by engineering the PbrR transcription factor to detect lead (Pb^{2+}) at environmentally relevant concentrations while minimizing cross-reactivity with zinc (Zn^{2+}). The study is well-designed, methodologically rigorous, and offers significant advancements in biosensor development. However, the performance of the ML model should be evaluated carefully as rational design was also used during the engineering of aTF. Also, following concerns should be addressed.

Line 213, 249, 275, how do you determine the number of mutants tested in each round?

Figure 5A. Error bar for some variants are quite large, for example Mutant8. Please explain.

Line 265, please explain the rational design process in detail.

Line 281, Again, how do you make the rational design? Rational design did produce improved variants, which seemed to compromise the effect of machine learning. Please clearly discuss the performance of ML model. Will these improved variants designed rationally be predicted by the ML model if enough sampling was used? Also, could the authors comment on the traditional ML model such as MLDE, and make a comparison with the one they used?

While the improved variants were obtained, the manuscript could better explain how specific mutations (e.g., D64K, P143R) contribute to the observed changes in biosensor behaviour. A discussion of structural or mechanistic insights (e.g., allosteric communication, DNA-binding dynamics) would strengthen the biological interpretation.

(Remarks on code availability)

Reviewer #3

(Remarks to the Author)

The manuscript by Wang et al. describes an integrated Design–Build–Test–Learn (DBTL) workflow that couples high-throughput cell-free expression with a multi-objective machine-learning (ML) approach to re-engineer the lead-responsive transcription factor PbrR.

By iterating five rounds—three ML-guided, two rational mutagenesis—the authors screened 2,024 variants (<1% of the single-mutation search space) and produced a hexamutant that (i) starts activating at $0.05 \mu\text{M Pb}^{2+}$, close to the proposed 10 ppb U.S. EPA action level, (ii) shows negligible response to $30 \mu\text{M Zn}^{2+}$, and (iii) retains activity after freeze-drying, suggesting possible point-of-use water tests.

The work is a follow-up to two previous papers, in particular Ika et al. (2024, ACS SynBio). In this earlier article, the authors

performed deep mutational scanning of PbrR and managed to decrease its limit of detection. However, this improvement came at the cost of increased promiscuity: engineered variants showed reduced selectivity, responding to other metals like cadmium, mercury, and zinc. In the current study, the authors use a dual-objective ML workflow to simultaneously optimize LOD and reduce cross-reactivity with zinc.

Overall, the study is timely, technically sound, and likely to interest readers working in synthetic biology, protein engineering, and on-site diagnostics.

However, there are some points that need to be addressed for the manuscript to be suitable for publication in Nature Communications.

1. I think more context should be given by the authors on their previous work in Ikas et al. Somehow, they managed to improve LOD to the legal limit, but selectivity got worse—hence the need for dual-objective optimization, for which ML seems well adapted. This would more clearly introduce the context of the present work and frame the problem.

2. I was surprised to see that the authors test only Zn²⁺. PbrR is well known to respond to other ions, such as cadmium, mercury, and copper. Mutations made by the authors could also affect, positively or negatively, binding to these ions. This effect was actually observed in Ikas (2024), where cross-reactivity increased significantly for several ions. Such cross-reactivity, especially if uncharacterized, could lead to false positives in field tests. Therefore, the authors *must* test the activity of their mutants with additional ions to validate selectivity for lead.

3. In the hypothesis that cross-reactivity exists, what would be the strategy? Can this framework be scaled to more than two parameters, simultaneously optimizing sensitivity and specificity for lead against multiple other ions?

4. The authors state: “The distribution of the mutations highlights the limitations of using rational engineering approaches for allosteric proteins as it is difficult to rationalize why mutations outside of the LBD would be important for tuning ligand selectivity.”

However, they performed such mutations in their previous paper and observed effects—they even found some of the same mutations. In that work, they wrote: “Most hotspots could not have been identified through structural analysis or canonical strategies alone. For example, most attempts to engineer a TF activity focus on the LBD; however, only one of our top single mutants from the site-saturation mutagenesis screen was a ligand-binding domain mutation.”

Therefore, it is not surprising that mutations in these domains appear again in the present study. This sentence and the associated message should be revised.

5. That being said, I would appreciate if the authors discussed in more detail the potential effects of the mutations outside the LBD. The fact that the method results in a mutant with “unexpected” mutations does not exempt the authors from trying to understand what is happening. Interestingly, this might provide insights into the regulation of transcription factors and how to better engineer them. Factors such as stability, affinity for DNA and allostery should be discussed. I agree that ML does not necessarily provide explainable results, but these results can still be used to learn something about the biology and engineering of these TFs.

6. Regarding the “diagnostics claims,” starting in the abstract: “We then show this engineered PbrR can be used in freeze-dried cell-free reactions as a portable diagnostic.” This is an overstatement. The authors freeze-dried the sensor, rehydrated it with laboratory water, and showed that it still responds to lead. To support claims of a diagnostic system, it would be necessary to test the sensor with real water samples to assess robustness under field-relevant conditions. Some of the authors have done this in previous work. Ideally, samples should be taken from known sites and independently analyzed for ion concentrations, then tested with the sensor. Even starting with municipal water would give the authors a first indication of whether the system is viable. This would help assess matrix effects and potential cross-reactivity that could limit field deployment and may highlight what remains to be done—hopefully, this will work!

(Remarks on code availability)

Reviewer #4

(Remarks to the Author)

General:

Wang et al. describe an ML based work flow to engineering the selectivity of a WT PbrR sensor that can detect Pb relative to Zn. After five rounds of ML they identify a mutant with 26 fold lower selectivity to Zn and that can detect 10 ppb of Pb more sensitively. The paper was a pleasure to read and the ML method used in particular the Multi-objective ICE to lower the selectivity to Zn was novel and interesting. Typically using other protein engineering methods e.g. continuous directed evolution to eliminate function such as response to a ligand is very difficult as counter selection is hard to do. Therefore ML methods such as the one developed by Wang et al. is valuable. Furthermore the data set generated is also very useful in advancing ML methods for biosensors. However, the manuscript could be improved significantly by addressing the following recommendations.

Major:

1) Novelty of the work. The way the story is written needs to be clarified. If the overall message is that the workflow or the method is the novelty of the work. Then applying this method on another sensor would be important to show. For example if the ML is the key advance then supporting this with data on fewer cycles required relative to other approaches like using

random library generation from top mutants or the method outlined in Nishikawa et al. 2024 Nat Comm (where FuncLib was used) could be important to emphasize. The key here is that higher order mutant combinations are easier to explore with the ML method. Of course, if the actual sensor is key, then testing it in real applications with different real waste water streams is important. Especially testing to see if there is any cross reactivity with other elements typically present in waste water.

2) ML for biosensors. The introduction could be modified to include additional ML studies that focus on the use of ML to design biosensors. For example papers such as DeepTFactor by Kim et al. PNAS 2020 , Zeng et al. Nat. Comm, 2024 for DL based DNA binding could be important to cite and compare with the current approach. There are many many ML methods for predicting TFs and binding sites and similarly there are many methods for predicting protein ligand interactions and binding that are relevant literature. Given all these existing methods it would be important to identify the key novelty for this paper. This reviewer would suggest that this would be a data driven iterative method for specifically engineering sequences of PbrR as opposed to some of those more general methods. The use of direction tokens and paired sequence variants appears similar to approaches in models like ProteinDPO, where pairs of sequences (one better than the other) were used to train the model. Therefore, it would be great if the author can explain the advantage and disadvantage of using such design of the Multi Objective ICE. The key value is the pair wise training which appears to be unique though this difference is a bit lost in this paper and should be highlighted more.

The other key piece is also the use of CFE system that can enable the generation of vast amount of data key for protein engineering. In fact, the ML methods should review other SOTA ML methods for protein engineering and place this in the appropriate context. The actual ML method used of course is not the highlight of this work rather the integration of ML methods, iterative cycles used and the CFE system to engineer selectivity would be the contribution.

3) Details of the training method and the dataset: The authors clearly explain how experimentally labeled mutants were converted to sequence pairs with direction token for ProtT5 model fine tuning. Some critical training details are missing. (1) The exact number of sequence pairs remained after filtering, which is the training data size. (2) Whether a train/validation split is used, the number of epochs or steps of training. (3) Quantitative metrics or other computational analysis on validation dataset (for example, perplexity) to show that the model can generate reliable novel sequences. Authors should include the dataset used in the training for Pb and Zn sensitivity in the github paper so that others in community can use this in future studies for example in training metal binding proteins. This step will enhance the value of this study as well. In the second round of training with FC and DR, was the same four objectives (+- Pb/+Zn) used for the DR ? Was this used in combination with FC which will lead to more cases right (16) ? Are there enough data points in each category ? Will there be a category that is not represented well ?

4) Distribution of mutations and allosteric proteins: The final 6 mutation sensor achieves its selectivity and sensitivity through mutations distributed across distinct functional domains: DBD, HTH motif, and LBD. The authors emphasize the limitations of using rational engineering approaches for allosteric proteins as it is difficult to rationalize why mutations outside of the LBD would be important for tuning ligand selectivity. The iterative hybrid design helped fill the gap by enabling the exploration and identification of non-typical residues critical for fine-tuning protein function that is otherwise difficult to predict. However, an elaboration of the specific ways in which their ML model along with rational design rounds specifically guided the discovery of said mutations, especially in domains other than the LBDs could help strengthen the overall narrative. Did initial ML predictions hint at these domains or was it rational design that helped primarily alongside ML detected residues? Did the ML models specifically learn the interactions between the mutations in the different domains? An interpretability analysis could provide valuable insights into how these complex relationships were captured and weighted and would help provide a mechanistic explanation.

5) Use of DR and FC in Round 2: What specific observation led to the conclusion that DR was a metric for improving selectivity over zinc compared to FC? Providing a direct comparison of how DR helped identify mutants would significantly strengthen this point.

Minor

- 1) In abstract both specificity and selectivity are discussed but not followed in the rest of the paper. What is the difference between specificity and selectivity ?
- 2) Github site does not seem to be populated ?
- 3) Details of the plasmids (pJL1) and constructs used in CFP are not sufficient especially details on the promoters and other parts of constructs expressing the TFs. Typically there is a list of plasmids generated at least the initial ones from which the library is based on.
- 4) How reproducible is the FC obtained when the library is screened repeatedly ?
- 5) The line "However, there were two mutants (D64K_N83F and N83I_K104V) that showed a higher fluorescent response to lead than zinc (Fig. S1)." Figure S1 shows D64K_N84F. This must be 83 ? Also Figure S2 x axis labels for round 0 do not seem to correspond to the top two mutants as the N83F mutant seems missing ??
- 6) Typo in the line "We tested high-order mutants because we observed improved sensor activity by combining multiple mutations in our previous PbrR engineering effort" Here the advantage of ML over other random mutations could be mentioned.
- 7) References 73 and 74 are not correct. I was expecting to the ProtT5 model and LoRA paper in this line "We use a transformer encoder-decoder model based on the T5 architecture, specifically the ProtT5-XL-UniRef50 model from Rostlab73"

(Remarks on code availability)

Github code was empty so could not evaluate the code. Happy to do this in the revised version.

Version 2:

Reviewer comments:

Reviewer #2

(Remarks to the Author)

The authors have addressed my concerns. The manuscript can be accepted now.

(Remarks on code availability)

Reviewer #3

(Remarks to the Author)

I thank the authors for providing additional explanations as well as additional data. They have mostly addressed my concerns.

I would just appreciate if the new increased cross-reactivity with Mercury be discussed-any hints about why this happens only with mercury?

I agree that given the different LOD, the sensor is still relevant for lead, but would like a bit more discussion on the potential impacts and implication of this new cross-reactivity generated by the workflow. For example: are they found commonly together in drinking water? I seems not from a brief survey. In what particular set ups this cross reactivity would be a problem?

Thank you.

(Remarks on code availability)

Reviewer #5

(Remarks to the Author)

I reviewed the revised manuscript and responses to previous reviewers and I believe the manuscript is adequately revised and ready for publication.

(Remarks on code availability)

Reviewer #2 (Remarks to the Author):

This manuscript presents a compelling integration of machine learning (ML) and cell-free gene expression (CFE) to engineer allosteric transcription factors (aTFs) for improved biosensor performance. The work addresses a critical challenge in synthetic biology: tuning multiple biosensor parameters (e.g., sensitivity and selectivity) simultaneously. The authors demonstrate the utility of their approach by engineering the PbrR transcription factor to detect lead (Pb^{2+}) at environmentally relevant concentrations while minimizing cross-reactivity with zinc (Zn^{2+}). The study is well-designed, methodologically rigorous, and offers significant advancements in biosensor development. However, the performance of the ML model should be evaluated carefully as rational design was also used during the engineering of aTF. Also, following concerns should be addressed.

We thank you for celebrating how our work addresses a critical challenge in synthetic biology and that the study is well designed, methodically rigorous, and provides a significant advance. We address your concerns below.

Line 213, 249, 275, how do you determine the number of mutants tested in each round?

In each round, we used a machine learning model to propose new mutant sequences, starting from a small set of "seed" mutants. These seeds are top performing sequences from previous experiments. Rather than picking just the single best, we select a few of the best to give the model multiple strong starting points. The model is formulated to propose new mutants that are predicted to perform better than the seed sequences based on the optimization objective (higher lead sensitivity and lower zinc sensitivity). The model generates 20 candidate sequences per seed. A parameter called "temperature" controls how creative or exploratory the model's suggestions are. Lower temperature values keep predictions closer to what the model has seen in experiments and higher values make the model propose more creative but less confident sequences. After generation, we remove previously tested sequences and prioritize sequences with higher order mutations for experimental validation. The final number of mutants tested in each round was constrained by well plate capacity and DNA cost, but mainly DNA synthesis costs since each construct was tested clonally. In the first round, we tested a larger number of mutants to generate sufficient data to train the model. In later rounds, we could test fewer mutants because the model had access to more data from previous rounds and could make more informed suggestions.

We add the following text to the revised manuscript as follows:

(Line 226) "With an ML model framework designed and trained on an initial dataset (Round 0), the first round of PbrR engineering towards increased lead sensitivity and decreased zinc selectivity consisted of 382 computationally predicted mutants ranging from 1st- to 6th-order mutants (i.e., amino acid changes). This number was chosen to match the capacity of the 384-well plate that is used in our assay, while leaving wells open for controls."

(Line 265) "In Round 2, we screened 200 computationally predicted higher-order mutants, with 100 mutants predicted from the model trained on normalized DR data at lead and zinc and the other 100 mutants predicted from the model trained on normalized DR data at zinc and normalized FC data at lead. As the model became better informed by data from earlier rounds, fewer mutants were needed to explore the sequence space effectively. Testing 100 mutants per metric allowed us to compare the performance of each training strategy while reducing DNA synthesis costs."

(Line 305) “Of the 108 mutants screened, 75 mutants had a normalized FC to lead greater than one and to zinc less than one (Fig. 4E) and 34 mutants had higher fluorescent signal to 1 mM lead than to 30 mM zinc (Fig. S1). We were again able to screen a reduced library size of 108 mutants because the large dataset from previous rounds increased the reliability of model predictions.”

Figure 5A. Error bar for some variants are quite large, for example Mutant 8. Please explain.

We note that the error bar for some variants in Figure 5A, such as Mutant 8, appear to be large. These variants are sensing lead and zinc concentrations that are below their respective limits of detection. Under these conditions, the biosensor remains essentially inactive, and the signals observed are primarily attributable to background noise rather than true detection.

Line 265, please explain the rational design process in detail.

We thank you for raising this question as it helped us realize we need to elaborate on our rational design process. In our previous PbrR engineering campaign (Ekas et al., 2024), we created 26 higher-order mutants, ranging from 2nd- to 5th-order, from 5 single mutations that each improved lead sensitivity. We observed the greatest improvement to lead sensitivity at the legal limit with a 4th-order mutant (M60L_P61N_D64K_G128I), and none of the 2nd- and 3rd-order mutants had the desired lead sensitivity. Based on this observation, we hypothesized that introducing a greater number of mutations to PbrR in this work would cause a more substantial change in its biosensor behavior to achieve our target sensitivity and selectivity. Therefore, when we observed overlap mutations at N83 and K104 in the three “winners” from Round 1 and 2, we hypothesized that these were key mutations for altering the sensitivity and selectivity of the biosensor and were likely needed in a higher-order mutant. To reduce DNA synthesis costs, we created a rational library (Round 2b) with all possible 4th- and 5th-order mutants that could be created from the six unique mutations identified in the two previous rounds.

Line 281, Again, how do you make the rational design? Rational design did produce improved variants, which seemed to compromise the effect of machine learning. Please clearly discussed the performance of ML model. Will these improved variants designed rationally be predicted by the ML model if enough sampling was used? Also, could the authors comment on the traditional ML model such as MLDE, and make a comparison with the one they used?

We appreciate the opportunity to clarify the relationship between the rationally designed variants and our machine learning guided approach.

The rational designs in our study are not independent of the ML model. Rather, they are informed combinations of high performing mutations that were either suggested by the model or already present in the initial experimental dataset. These combinations were generated manually after ML rounds and are composed of mutations at sites that the model has already encountered during training, so the rational variants do not introduce new mutational sites outside the model’s training distribution.

Because the rational designs were tested immediately following an ML round, the model had not yet seen the outcomes of these combinations during training. However, if these variants had instead been included in the next round of ML training, the model would have incorporated this validated information and adjusted its learned probabilities accordingly. Given the stochastic nature of the sequence-to-sequence model, this would have substantially increased the likelihood of it proposing these same combinations in

its own predictions. However, we are still limited in generating a large dataset representing a larger fraction of the potential sequence space for our protein because of DNA synthesis costs. Currently, 500 PbrR mutants cost about \$19K. If DNA synthesis could be decreased by 10-100x, we could significantly increase our library size and improve ML training.

In this sense, the rational design phase acts as a valuable form of targeted exploitation, strategically recombining mutations that the model has already identified as promising. We believe that the most effective search strategies balance exploration (broadly surveying the landscape) and exploitation (focusing on known beneficial regions). By explicitly incorporating this balance, the rational design round complements the ML-driven exploration, creating a synergy that uses both data driven discovery and targeted improvements.

Regarding the performance of the ML model, it improves over successive rounds and is capable of generalizing beyond training distributions. While rational design appears to outperform the model in several cases, this is expected since rational designs are built on top of model suggested parts, not independent hypotheses. In fact, the rational variants serve to validate the model's predictions and provide a strong foundation for the next round of retraining. For example, the model proposed mutants with P104R mutation, which alone is not a good candidate but resulted in good higher order mutation.

We discuss these new points in the revised manuscript:

(Line 272) “Overall, we observed a modest increase in the number of mutants with a higher normalized FC to lead relative to zinc (Fig. 4C). However, most mutants still showed a stronger signal response to zinc than lead (Fig. S1). Mutants predicted from the model trained on lead normalized FC and zinc normalized DR generally had higher leak. Despite this, the ML model began proposing hits with higher order mutations that were not obvious or additive. For example, the model predicted the mutant N83I_K104V_H106A_P143R, which included the H106A mutation, a substitution not previously shown to increase lead sensitivity on its own. This mutant displayed higher signal to lead than zinc and was validated in a by-hand experiment (Fig. S2).”

(Line 285) “We noted that the three “winners” from Round 1 and 2 contained overlap mutations at N83 and K104 and decided to perform a round of rational engineering, Round 2b, by creating ten higher order (4th- and 5th-order) mutants from the six unique mutations at five residue positions from previous winners, consisting of K64D, N83I or N83F, K104V, H106A, and P143R (Fig. 4D). These designs were informed directly by prior ML-guided rounds and experimental validation, reflecting a strategic recombination of high performing substitutions. While these rational variants were not proposed de novo by the model, they drew directly from the higher order combinations that the model has already prioritized, which is a targeted exploitation of model discovered signals. Targeted exploitation of high performing mutants through combination has been observed to improve sensor activity by combining multiple mutations in our previous PbrR engineering efforts¹.”

(Line 331) “The top mutants of Round 3b exhibited high sensitivity to lead at the EPA action level without activation by zinc, indicating low likelihood of false positive results to due zinc crosstalk. Importantly, the ML model and rational design served complementary roles throughout the engineering process. Rational design efficiently built on validated mutations by combining previously successful substitutions into higher order mutants, while the model proposed unexpected non-additive combinations where some individual mutations alone offered little benefit. When rationally designed mutants were incorporated into subsequent ML training, they

strengthened the model's confidence in key regions of the sequence space and improved its ability to prioritize synergistic mutations. This exchange between computationally proposed and experimentally guided recombination enabled more focused searches and was effective in realizing the final design."

Finally, when comparing our model to traditional MLDE methods, several key differences stand out. Our approach leverages sequence pairs labeled with the direction of functional change between mutants, enabling the model to learn from both positive and negative outcomes. It inherently supports multi-objective optimization via multiple directional tokens, eliminating the need to train separate models for each objective. Unlike many MLDE methods that depend on scalar predictors, which are often unreliable in data-scarce scenarios, our model uses qualitative comparisons instead of explicit regression. Many existing MLDE approaches rely on predictors that score individual mutations or precomposed libraries for screening²⁻⁸.

A recent work, ProteinDPO⁹, uses a similar framework but lacks multiple directional tokens necessary for multi-objective optimization and does not incorporate bidirectional information such as how to improve or worsen function according to the objective. Incorporating negative examples is crucial for refining models and guiding them on what outputs to avoid¹⁰. Importantly, ProteinDPO focuses solely on stability and does not address multi-objective tasks like ligand selectivity, underscoring the advantage of our approach in handling more complex optimization scenarios.

We have revised the introduction section to better position our method in the current literature:

"Machine learning (ML)-guided directed evolution has transformed the way we navigate vast protein sequence-function landscapes, making exploration faster and less reliant on exhaustive experiments^{3-8,11-13}. At its core, ML supports protein engineering in two complementary ways: predictive models, which score given sequences or structures, and generative models, which propose new ones.

Predictive models act as evaluators to screen predefined libraries and prioritize candidates for experimental validation. Examples include zero-shot predictors, which infer fitness directly from evolutionary or structural context using pretrained protein large language models (pLLMs)¹⁴⁻²⁰, and supervised models trained on sequence-function datasets to guide optimization for specific tasks such as catalysis^{17,21-24}. Classification-based models, like DeepTFactor²⁵ (which identifies transcription factors) and ESM-DBP²⁶ (which predicts DNA-binding proteins), represent another subset. These models excel at discovery and annotation tasks but mainly identify candidate scaffolds rather than optimize functional properties. Traditional ML-directed evolution (MLDE) approaches can be used to combine predictive models with manually designed libraries²⁻⁸, making results sensitive to predictor accuracy, prior assumptions, and biases in the training data.

Generative models, by contrast, create novel sequences or structures tailored to specific objectives. Examples include structure-generating models such as RFdiffusion^{27,28}, which create protein backbones that scaffold functional motifs, and mutation proposing models such as FuncLib²⁹, which suggest active-site substitutions without manual library construction. However, such models have limitations: they often depend on high-quality structural information and abundant sequence homologs, and may fail for dynamic or allosterically regulated proteins. More recent approaches, such as preference-based learning (e.g., ProteinDPO⁹), incorporate relative comparisons between variants, but their scope is narrow—focusing on single objectives and lacking adaptability for tasks where tradeoffs between properties must be considered. This is especially relevant for biosensors where increasing sensitivity often reduces selectivity.

To overcome existing limitations, we set out to develop a directional, multi-objective ML model for engineering aTF-based biosensors that relies on a controlled extrapolation framework (Fig. 1)³⁰. This model uses a sequence-to-sequence architecture to learn how amino acid sequence changes influence protein function, guided by tokens that encode the direction of property change. This approach eliminates the need for downstream predictors and allows direct manipulation of the model's latent space. Unlike preference-based models that only highlight improvements, our directional token approach exposes the model to both beneficial and detrimental mutations within the same training framework. For example, when the model sees a sequence pair labeled with 'decrease/increase' tokens, it simultaneously learns which mutations harm the first property while benefiting the second. This bidirectional learning provides a rich training signal about the mutational landscape¹⁰."

While the improved variants were obtained, the manuscript could better explain how specific mutations (e.g., D64K, P143R) contribute to the observed changes in biosensor behaviour. A discussion of structural or mechanistic insights (e.g., allosteric communication, DNA-binding dynamics) would strengthen the biological interpretation.

We share the reviewer's excitement to better understand the biophysics of the engineered aTF. We are especially interested in this because there are some unusual mutations (e.g., proline to arginine), and mechanistic insights would strengthen the biological interpretation.

We speculate that the observed shift in sensitivity toward lead over zinc may result from a combination of local changes in the ligand-binding site and long-range allosteric effects because the final mutant has mutations in different regions of the protein. Lead (II) typically has more flexible coordinate geometries compared to zinc(II), which prefers tetrahedral coordination. Mutations in the ligand-binding domain (K104T, H106A, and P143R) may subtly reshape the binding pocket's geometry or electrostatics to disfavor zinc coordination. Mutations in the DNA-binding domain (D64K) and the helix-turn-helix motif (N83I) may influence allosteric communication between domains by altering local flexibility or packing. For example, replacing the polar asparagine with the smaller, hydrophobic isoleucine in the HTH region could affect helix stability or dynamics, potentially altering the sensor's conformational response to different metals.

The following text has been added to the revised manuscript:

(Line 430) "The distribution of the mutations highlights the limitations of using rational engineering approaches for allosteric proteins as it is difficult to rationalize why this specific combination of mutations would be important for tuning ligand selectivity. We hypothesize that these mutations are impacting metal ion coordination, DNA affinity, and allostery. For example, Pb²⁺ typically has more flexible coordinate geometries with proteins compared to zinc²⁺, which prefers tetrahedral coordination^{31,32}. Mutations in the ligand-binding domain may subtly reshape the binding pocket's geometry or electrostatics to disfavor zinc coordination. Additionally, mutations in the HTH motif may influence the allosteric communication between the LBD and DBD domains, altering the transcriptional response of the biosensor to different metals³³."

Reviewer #3 (Remarks to the Author):

The manuscript by Wang et al. describes an integrated Design–Build–Test–Learn (DBTL) workflow that couples high-throughput cell-free expression with a multi-objective machine-learning (ML) approach to re-engineer the lead-responsive transcription factor PbrR.

By iterating five rounds—three ML-guided, two rational mutagenesis—the authors screened 2,024 variants (<1% of the single-mutation search space) and produced a hexamutant that (i) starts activating at 0.05 μM Pb^{2+} , close to the proposed 10 ppb U.S. EPA action level, (ii) shows negligible response to 30 μM Zn^{2+} , and (iii) retains activity after freeze-drying, suggesting possible point-of-use water tests.

The work is a follow-up to two previous papers, in particular Ikas et al. (2024, ACS SynBio). In this earlier article, the authors performed deep mutational scanning of PbrR and managed to decrease its limit of detection. However, this improvement came at the cost of increased promiscuity: engineered variants showed reduced selectivity, responding to other metals like cadmium, mercury, and zinc. In the current study, the authors use a dual-objective ML workflow to simultaneously optimize LOD and reduce cross-reactivity with zinc.

Overall, the study is timely, technically sound, and likely to interest readers working in synthetic biology, protein engineering, and on-site diagnostics. However, there are some points that need to be addressed for the manuscript to be suitable for publication in Nature Communications.

We appreciate you celebrating our story as timely, technically sound, interesting and suitable for *Nature Communications*.

1. I think more context should be given by the authors on their previous work in Ikas et al. Somehow, they managed to improve LOD to the legal limit, but selectivity got worse—hence the need for dual-objective optimization, for which ML seems well adapted. This would more clearly introduce the context of the present work and frame the problem.

We now provide more context on the need for dual-objective optimization given the results of our previous work.

We add the following lines:

*(Line 115) “We implemented an active learning framework combining this directional ML model with a cell-free expression (CFE) system. The CFE system uses crude cellular extracts and reaction components (e.g., energy substrates, amino acids) to enable high-throughput transcription and translation of DNA templates outside living cells³⁴⁻³⁶. Building on our previous work^{1,37}, we integrated ML-guided design to achieve multi-objective optimization of the lead-responsive aTF PbrR, originally from the megaplasmid pMOL30 of *Cupriavidus metallidurans*³⁸. Lead was selected because of its severe public health impact³⁹⁻⁴¹. In the United States alone, there are an estimated 9.2 million lead service lines still in use⁴². Using the open, scalable CFE system, we rapidly generated positive and negative sequence-function data across multiple ligand conditions to train the model and iteratively refine predictions. This integrated, data-driven workflow enabled engineering of PbrR variants that balance sensitivity and selectivity, meeting performance requirements for lead detection in drinking water.”*

2. I was surprised to see that the authors test only Zn^{2+} . PbrR is well known to respond to other ions, such as cadmium, mercury, and copper. Mutations made by the authors could also affect, positively or negatively, binding to these ions. This effect was actually observed in Ikas (2024), where cross-reactivity increased significantly for several ions. Such cross-reactivity, especially if uncharacterized, could lead to false positives in field tests. Therefore, the authors *must* test the activity of their mutants with additional ions to validate selectivity for lead.

We agree with you and have now generated these data, which are found in the supplement (**Fig. S4**). We tested wildtype and our best mutant (D64K_N83I_I90A_K104T_H106A_P143R) against 7 other ions, including cadmium, mercury, and copper, at concentrations of 1 μ M, 10 μ M, and 100 μ M. (Mercury at 100 μ M was not tested due to its high toxicity). We observed that the best mutant is only cross-reactive to mercury and is still highly selective for lead compared to previously engineered PbrR mutants. In field tests, a positive result due to mercury contamination in the water would still be beneficial given its toxicity.

3. In the hypothesis that cross-reactivity exists, what would be the strategy? Can this framework be scaled to more than two parameters, simultaneously optimizing sensitivity and specificity for lead against multiple other ions?

We think that this can be scaled to more parameters. A great advantage of cell-free expression (CFE) in well plates is that multiple parameters can be tested simultaneously. By using an Echo robot, we can easily test a library of mutants against another ion by dispensing the mutant DNA library into a well plate containing CFE reaction components and the ligand of interest³⁷. This offers an advantage to whole cell aTF biosensor engineering as cell-free systems allows us to test ligands that may be toxic to living cells or may be blocked by the cell membrane. Additionally, our assay can be scaled to use 1536-well plates to test more parameters. We need lower DNA synthesis costs to test larger libraries of combinatorial mutants to improve ML training.

The machine learning component of our framework is designed to handle multiple objectives and can be extended to optimize for more than two parameters simultaneously. However, as the number of objectives grows, the complexity of the optimization task increases. The model will learn more refined representations of the training data if it adequately represents all combinations of objectives. For example, if two objectives each have two directions (increase or decrease), this results in four possible categories. With more objectives, the number of such categories grows, and imbalanced representation among them can lead to biased model predictions. While the model is trainable under such conditions, performance may potentially be affected. We now discuss this limitation in the Lines 230-236.

4. The authors state: “The distribution of the mutations highlights the limitations of using rational engineering approaches for allosteric proteins as it is difficult to rationalize why mutations outside of the LBD would be important for tuning ligand selectivity.”

However, they performed such mutations in their previous paper and observed effects—they even found some of the same mutations. In that work, they wrote: “Most hotspots could not have been identified through structural analysis or canonical strategies alone. For example, most attempts to engineer aTF activity focus on the LBD; however, only one of our top single mutants from the site-saturation mutagenesis screen was a ligand-binding domain mutation.”

Therefore, it is not surprising that mutations in these domains appear again in the present study. This sentence and the associated message should be revised.

We appreciate the reviewer pointing out how our statement is confusing. We have revised it as follows:

(Line 430) “The distribution of the mutations highlights the limitations of using rational engineering approaches for allosteric proteins as it is difficult to rationalize why this specific combination of mutations would be important for tuning ligand selectivity.”

5. That being said, I would appreciate if the authors discussed in more detail the potential effects of the mutations outside the LBD. The fact that the method results in a mutant with “unexpected” mutations does not exempt the authors from trying to understand what is happening. Interestingly, this might provide insights into the regulation of transcription factors and how to better engineer them. Factors such as stability, affinity for DNA and allostery should be discussed. I agree that ML does not necessarily provide explainable results, but these results can still be used to learn something about the biology and engineering of these TFs.

We are also interested in understanding how our mutations are impacting the biology of PbrR, such as stability, affinity for DNA, and allostery. We hypothesize that the sensitivity and selectivity shifts of our best mutant result from both local changes in the LBD and long-range allosteric effects. For example, Pb(II) coordination in proteins is more flexible than Zn(II) coordination, which prefers tetrahedral coordination. Mutations K104T, H106A, and P143R in the LBD may subtly reshape the geometry and electrostatics of the ligand binding pocket away from tetrahedral coordination, allow for lead selectivity over zinc^{31,32}. Additionally, we hypothesize that mutations in the HTH motif alter the allosteric communication between the LBD and DBD domains, altering the transcriptional response of the biosensor to different metals³³.

We add the following text to the revised manuscript:

(Line 432) “We hypothesize that these mutations are impacting metal ion coordination, DNA affinity, and allostery. For example, Pb²⁺ typically has more flexible coordination geometries with proteins compared to zinc²⁺, which prefers tetrahedral coordination^{31,32}. Mutations in the ligand-binding domain may subtly reshape the binding pocket’s geometry or electrostatics to disfavor zinc coordination. Additionally, mutations in the HTH motif may influence the allosteric communication between the LBD and DBD domains, altering the transcriptional response of the biosensor to different metals³³.”

6. Regarding the “diagnostics claims,” starting in the abstract: “We then show this engineered PbrR can be used in freeze-dried cell-free reactions as a portable diagnostic.” This is an overstatement. The authors freeze-dried the sensor, rehydrated it with laboratory water, and showed that it still responds to lead. To support claims of a diagnostic system, it would be necessary to test the sensor with real water samples to assess robustness under field-relevant conditions. Some of the authors have done this in previous work. Ideally, samples should be taken from known sites and independently analyzed for ion concentrations, then tested with the sensor. Even starting with municipal water would give the authors a first indication of whether the system is viable. This would help assess matrix effects and potential cross-reactivity that could limit field deployment and may highlight what remains to be done—hopefully, this will work!

We agree that it is important to test the sensor with real water samples to assess robustness under field-relevant conditions. In the revised manuscript, we collaborated with the Gaillard and Lucks Labs at Northwestern to test municipal water samples from homes in the Chicagoland area. To test these samples, we modified the reporter system of our biosensor to use the C23DO enzyme to cleave colorless catechol into the yellow pigment 2-hydroxymuconate semialdehyde. We show these data in **Figure 6G**. This has resulted in the addition of multiple authors on the paper.

Reviewer #4 (Remarks to the Author):

General:

Wang et al. describe an ML based work flow to engineering the selectivity of a WT PbrR sensor that can detect Pb relative to Zn. After five rounds of ML they identify a mutant with 26 fold lower selectivity to Zn and that can detect 10 ppb of Pb more sensitively. The paper was a pleasure to read and the ML method used in particular the Multi-objective ICE to lower the selectivity to Zn was novel and interesting. Typically using other protein engineering methods e.g. continuous directed evolution to eliminate function such as response to a ligand is very difficult as counter selection is hard to do. Therefore ML methods such as the one developed by Wang et al. is valuable. Furthermore the data set generated is also very useful in advancing ML methods for biosensors. However, the manuscript could be improved significantly by addressing the following recommendations.

Thank you highlighting that the paper was a pleasure to read and novel, both the method and the dataset.

Major:

1) Novelty of the work. The way the story is written needs to be clarified. If the overall message is that the workflow or the method is the novelty of the work. Then applying this method on another sensor would be important to show. For example if the ML is the key advance then supporting this with data on fewer cycles required relative to other approaches like using random library generation from top mutants or the method outlined in Nishikawa et al. 2024 Nat Comm (where FuncLib was used) could be important to emphasize. The key here is that higher order mutant combinations are easier to explore with the ML method. Of course, if the actual sensor is key, then testing it in real applications with different real waste water streams is important. Especially testing to see if there is any cross reactivity with other elements typically present in waste water.

We agree with you that the story walks the line between two innovations - the ML model and the engineered biosensor. We believe both are important, not one over the other. The strength of our work lies in the integration of two innovations as we apply our ML model to a real-world problem and only achieve dual-optimization and application requirements because of the ML model and cell-free screening assay. The ML model enables efficient exploration of higher-order mutant combinations that would be impractical to screen exhaustively through random library generation or traditional computational methods.

In the revised manuscript, we have expanded on our explanation of the ML model in the introduction, and we have provided new data testing our biosensor on municipal water samples (Fig. 6G).

2) ML for biosensors. The introduction could be modified to include additional ML studies that focus on the use of ML to design biosensors. For example papers such as DeepTFactor by Kim et al. PNAS 2020 , Zeng et al. Nat. Comm, 2024 for DL based DNA binding could be important to cite and compare with the current approach. There are many many ML methods for predicting TFs and binding sites and similarly there are many methods for predicting protein ligand interactions and binding that are relevant literature. Given all these existing methods it would be important to identify the key novelty for this paper. This reviewer would suggest that this would be a data driven iterative method for specifically engineering sequences of PbrR as opposed to some of those more general methods. The use of direction tokens and paired sequence variants appears similar to approaches in models like ProteinDPO, where pairs of sequences (one better than the other) were used to train the model. Therefore, it would be great if the author can explain the advantage and disadvantage of using such design of the Multi Objective ICE. The

key value is the pair wise training which appears to be unique though this difference is a bit lost in this paper and should be highlighted more.

The other key piece is also the use of CFE system that can enable the generation of vast amount of data key for protein engineering. In fact, the ML methods should review other SOTA ML methods for protein engineering and place this in the appropriate context. The actual ML method used of course is not the highlight of this work rather the integration of ML methods, iterative cycles used and the CFE system to engineer selectivity would be the contribution.

We thank you for this thorough and thoughtful comment. We fully agree that it is important to position our work in the broader context of existing ML approaches for transcription factor (TF) engineering, protein–ligand interaction prediction, and general protein optimization. In the revised manuscript, we expand the Introduction and Discussion to include and compare relevant literature, including DeepTFactor and ESM-DBP, which use deep learning to predict TF activity and DNA binding, respectively.

While these methods are powerful in identifying candidate TFs or predicting properties from sequence, their primary focus is classification or prediction, rather than iterative sequence design. In contrast, our work focuses specifically on engineering a known TF biosensor (PbrR) to optimize functional performance across multiple objectives. Our contribution lies not in general TF prediction, but in developing a data driven, sequence-to-sequence ML framework tailored to multi-objective biosensor engineering, and in demonstrating its integration with a high-throughput cell free screening platform.

We appreciate you pointing out the relevance of paired sequence-based approaches like ProteinDPO. While our Multi-Objective ICE framework shares some conceptual ground with preference-based methods, it differs critically in its use of directional tokens and bidirectional training. Unlike methods that only show the model how to improve sequences, our directional token approach exposes the model to both beneficial and detrimental mutations within the same training framework. When the model sees a sequence pair labeled with “decrease/increase” tokens, it learns which edits harm one property while benefiting another. This bidirectional learning provides a richer training signal about the fitness landscape and helps the model develop a more complete understanding of both desirable and undesirable mutational directions.

This design aligns with well-established principles in machine learning, particularly in contrastive learning and representation learning, where exposure to both positive and negative examples enhances generalization and prevents overfitting^{10,43}.

In the context of protein design, this means our model learns not just where to go (toward higher fitness), but also where not to go, helping it avoid local optima in complex multi-objective spaces. Each sequence pair effectively doubles the information content compared to unidirectional methods, which is especially valuable in data limited regimes.

In addition, we agree with the reviewer that the integration of ML with a cell-free expression (CFE) platform is a critical strength of our framework. We now clarify this synergy between computational modeling and experimental throughput as a core contribution of our work.

Finally, we agree with your assessment that the novelty of our work is not solely in the ML architecture, but in how the ML framework is specialized for biosensor engineering, supports multi-objective optimization, and is tightly coupled to experimental feedback via CFE. We believe this integration and the

efficient exploration of higher-order mutant combinations it enables provides a meaningful advance beyond existing approaches such as FuncLib and we thank you for highlighting this point.

We have revised the manuscript to clarify this distinction and will cite supporting ML literature:

(Line 74) “Machine learning (ML)-guided directed evolution has transformed the way we navigate vast protein sequence-function landscapes, making exploration faster and less reliant on exhaustive experiments^{3-8,11-13}. At its core, ML supports protein engineering in two complementary ways: predictive models, which score given sequences or structures, and generative models, which propose new ones.

Predictive models act as evaluators to screen predefined libraries and prioritize candidates for experimental validation. Examples include zero-shot predictors, which infer fitness directly from evolutionary or structural context using pretrained protein large language models (pLLMs)¹⁴⁻²⁰, and supervised models trained on sequence-function datasets to guide optimization for specific tasks such as catalysis^{17,21-24}. Classification-based models, like DeepTFactor²⁵ (which identifies transcription factors) and ESM-DBP²⁶ (which predicts DNA-binding proteins), represent another subset. These models excel at discovery and annotation tasks but mainly identify candidate scaffolds rather than optimize functional properties. Traditional ML-directed evolution (MLDE) approaches can be used to combine predictive models with manually designed libraries²⁻⁸, making results sensitive to predictor accuracy, prior assumptions, and biases in the training data.

Generative models, by contrast, create novel sequences or structures tailored to specific objectives. Examples include structure-generating models such as RFdiffusion^{27,28}, which create protein backbones that scaffold functional motifs, and mutation proposing models such as FuncLib²⁹, which suggest active-site substitutions without manual library construction. However, such models have limitations: they often depend on high-quality structural information and abundant sequence homologs, and may fail for dynamic or allosterically regulated proteins. More recent approaches, such as preference-based learning (e.g., ProteinDPO⁹), incorporate relative comparisons between variants, but their scope is narrow—focusing on single objectives and lacking adaptability for tasks where tradeoffs between properties must be considered. This is especially relevant for biosensors where increasing sensitivity often reduces selectivity.

*To overcome existing limitations, we set out to develop a directional, multi-objective ML model for engineering aTF-based biosensors that relies on a controlled extrapolation framework (**Fig. 1**)³⁰. This model uses a sequence-to-sequence architecture to learn how amino acid sequence changes influence protein function, guided by tokens that encode the direction of property change. This approach eliminates the need for downstream predictors and allows direct manipulation of the model’s latent space. Unlike preference-based models that only highlight improvements, our directional token approach exposes the model to both beneficial and detrimental mutations within the same training framework. For example, when the model sees a sequence pair labeled with ‘decrease/increase’ tokens, it simultaneously learns which mutations harm the first property while benefiting the second. This bidirectional learning provides a rich training signal about the mutational landscape¹⁰.*

We implemented an active learning framework combining this directional ML model with a cell-free expression (CFE) system. The CFE system uses crude cellular extracts and reaction components (e.g., energy substrates, amino acids) to enable high-throughput transcription and translation of DNA templates outside living cells³⁴⁻³⁶. Building on our previous work^{1,37}, we integrated ML-guided design to achieve multi-objective optimization of the lead-responsive aTF

PbrR, originally from the megaplasmid pMOL30 of Cupriavidus metallidurans³⁸. Lead was selected because of its severe public health impact³⁹⁻⁴¹. In the United States alone, there are an estimated 9.2 million lead service lines still in use⁴². Using the open, scalable CFE system, we rapidly generated positive and negative sequence-function data across multiple ligand conditions to train the model and iteratively refine predictions. This integrated, data-driven workflow enabled engineering of PbrR variants that balance sensitivity and selectivity, meeting performance requirements for lead detection in drinking water.”

3) Details of the training method and the dataset: The authors clearly explain how experimentally labeled mutants were converted to sequence pairs with direction token for ProtT5 model fine tuning. Some critical training details are missing. (1) The exact number of sequence pairs remained after filtering, which is the training data size. (2) Whether a train/validation split is used, the number of epochs or steps of training. (3) Quantitative metrics or other computational analysis on validation dataset (for example, perplexity) to show that the model can generate reliable novel sequences. Authors should include the dataset used in the training for Pb and Zn sensitivity in the github paper so that others in community can use this in future studies for example in training metal binding proteins. This step will enhance the value of this study as well. In the second round of training with FC and DR, was the same four objectives (+- Pb/+Zn) used for the DR? Was this used in combination with FC which will lead to more cases right (16)? Are there enough data points in each category? Will there be a category that is not represented well?

We thank you for raising these important points. Below, we address each aspect in detail.

(1) Training dataset size

In each round, we used our ML model to propose new mutant sequences starting from a small set of high performing "seed" variants, typically multiple top performers rather than just the global best to allow the model to generalize from multiple strong baselines. The model is explicitly formulated to generate sequences predicted to outperform these seeds in the desired direction (e.g., higher Pb sensitivity and lower Zn sensitivity). For each seed, the model generates 20 candidate sequences. A "temperature" parameter modulates the diversity of generated sequences (lower values produce conservative edits close to prior observations, while higher values produce more exploratory mutations). We prioritize mutants with higher-order edits not previously tested and remove duplicates from earlier rounds. The number of mutants tested per round is constrained by experimental capacity (384-well plates) and budget.

The approximate size of the training dataset after filtering was on the order of 1e4 sequence pairs per round. In early rounds, more variants were tested to establish a diverse training base. In later rounds, fewer new variants were required as the model had access to a growing history of labeled data, allowing more informed extrapolation.

We add these lines to the revised manuscript:

(Line 226) “With an ML model framework designed and trained on an initial dataset (Round 0), the first round of PbrR engineering towards increased lead sensitivity and decreased zinc selectivity consisted of 382 computationally predicted mutants ranging from 1st- to 6th-order mutants (i.e., amino acid changes). This number was chosen to match the capacity of the 384-well plate that is used in our assay, while leaving wells open for controls.”

(Line 265) “In Round 2, we screened 200 computationally predicted higher-order mutants, with 100 mutants predicted from the model trained on normalized DR data at lead and zinc and the other 100 mutants predicted from the model trained on normalized DR data at zinc and

normalized FC data at lead. As the model became better informed by data from earlier rounds, fewer mutants were needed to explore the sequence space effectively. Testing 100 mutants per metric allowed us to compare the performance of each training strategy while reducing DNA synthesis costs.”

(Line 305) “Of the 108 mutants screened, 75 mutants had a normalized FC to lead greater than one and to zinc less than one (Fig. 4E) and 34 mutants had higher fluorescent signal to 1 mM lead than to 30 mM zinc (Fig. S1). We were again able to screen a reduced library size of 108 mutants because the large dataset from previous rounds increased the reliability of model predictions.”

(2) Training procedure, validation, and evaluation

Conventional train/validation/test splits are not directly applicable to our iterative, active-learning framework. Because our model is generative and designed to extrapolate beyond the current training distribution, it is trained on all available labeled sequence pairs in each round to maximize learning from the known mutational landscape.

Rather than holding out a fixed validation set, we evaluate model performance through real-world experimental validation in each design cycle. New sequences are proposed, tested *in vitro*, and the results fed back into the model in the next round, mirroring a closed-loop design-build-test-learn cycle. This approach prioritizes generalization to novel, higher-order combinations not observed in training rather than optimizing predictive accuracy on a fixed test set.

We train for a single epoch per round to avoid overfitting, as the dataset typically contains sufficient diversity. The parameters including the number of epochs used in training is provided in the supplementary information (Table S1).

(Line 542) “We use top-k sampling ($k = 10$) to generate diverse candidates. For each seed sequence, 20 candidates were proposed, and we filter by edit distance or number of mutations from the wildtype sequence as needed.”

(3) Evaluation metrics and dataset availability

To assess whether the model is generating valid sequences consistent with the training distribution, we use SacreBLEUScore, a BLEU-style metric from the original ICE framework, which compares generated sequences to ground-truth edits in a seq2seq context. While this provides a proxy for sequence quality, we also emphasize that perfect sequence match is not necessary, since alternative mutations may still yield desirable phenotypes due to epistasis. Therefore, our ultimate evaluation metric is experimental performance.

We agree with you that community access to the data is important. In our GitHub repository, we provide a quickstart guide with an example subset of our dataset (input.csv) to enable other researchers to generate novel sequences using our model. Upon publication, we will make the full Pb/Zn sensitivity dataset used for training publicly available via the repository.

The link to the Github page is: https://github.com/ShuklaGroup/multiobjective_controlled_extrapolation

The link to the input data for running the example under quickstart on Github is: <https://uofi.box.com/s/qpaaft9ge9f3ofqq7aybqwkxkid94fgd>

They can both be found under the code availability section.

(Line 524) “Loss is computed via token-level cross-entropy, and the model is optimized using AdamW with a learning rate of 1e-4 and weight decay of 1e-4, using Hugging Face’s Seq2SeqTrainer. During training, to assess whether the model is generating valid sequences consistent with the training distribution, we use the SacreBLEU score⁴⁴.”

(4) Multi-objective structure and data balancing

The FC and DR data both capture regulatory behavior with respect to Pb and Zn but represent different experimental measurements. We train separate models for FC and DR datasets to respect these distinctions while using the same four directional objectives (+Pb, -Pb, +Zn, -Zn). We ensure that the dataset is balanced across these objectives, and each category is sufficiently represented to provide meaningful supervision. While highly selective variants (e.g., high Pb/low Zn) are rarer due to the correlated nature of TF response, our training strategy relies on pairwise comparisons, not absolute fitness. This enables robust learning even when raw values are not extreme, as long as comparative ordering between sequences is preserved. We explain this in the methods section (Lines 200-206).

We appreciate the reviewer’s insightful questions and agree that sharing the dataset will significantly enhance the utility of our study to the broader protein engineering community. We want others to be able to build off our work.

4) Distribution of mutations and allosteric proteins: The final 6 mutation sensor achieves its selectivity and sensitivity through mutations distributed across distinct functional domains: DBD, HTH motif, and LBD. The authors emphasize the limitations of using rational engineering approaches for allosteric proteins as it is difficult to rationalize why mutations outside of the LBD would be important for tuning ligand selectivity. The iterative hybrid design helped fill the gap by enabling the exploration and identification of non-typical residues critical for fine-tuning protein function that is otherwise difficult to predict. However, an elaboration of the specific ways in which their ML model along with rational design rounds specifically guided the discovery of said mutations, especially in domains other than the LBDs could help strengthen the overall narrative. Did initial ML predictions hint at these domains or was it rational design that helped primarily alongside ML detected residues? Did the ML models specifically learn the interactions between the mutations in the different domains? An interpretability analysis could provide valuable insights into how these complex relationships were captured and weighted and would help provide a mechanistic explanation.

We thank you for this insightful comment. Indeed, the final six-mutation transcription factor mutants derive function from mutations distributed across the DBD, HTH motif, and LBD, highlighting the challenge of rationally engineering allosteric proteins. Our iterative design framework explicitly addresses this challenge by integrating rational rounds with ML-guided exploration. In every ML-guided round, the model proposed higher-order mutants spanning multiple domains, suggesting that it had learned the importance of cross-domain interactions. Rational design rounds then built upon these cross-domain mutations, refining combinations that were both functional and selective.

While our current model architecture does not lend itself easily to interpretability, given the embeddings don’t correspond cleanly to interpretable biophysical features so common interpretability techniques that highlight specific input features will not yield clear biological insights in our case. However, the model’s output probabilities consistently prioritized cross-domain mutants, providing functional evidence that the model had implicitly learned that edits in multiple domains were important for improving performance.

We agree that future iterations of this ML framework would benefit from enhanced interpretability, potentially through modifications to the model architecture or training objective to encourage mechanistic insight alongside predictive performance. This is an exciting direction we plan to pursue.

5) Use of DR and FC in Round 2: What specific observation led to the conclusion that DR was a metric for improving selectivity over zinc compared to FC? Providing a direct comparison of how DR helped identify mutants would significantly strengthen this point.

There are three main biosensor characteristics that are general targets for engineering: (1) sensitivity/selectivity, (2) signal response, and (3) kinetics. While FC captures sensitivity/selectivity, we were motivated to include DR as a metric for ML training to improve the signal response of the mutants. In Round 1, we observed that mutants with lead selectivity over zinc had very low fluorescent signal ($\sim 0.1 \mu\text{M}$ FITC at $1 \mu\text{M}$ Pb). While the difference in signal between ligand conditions is distinguishable using a plate reader, we were concerned about distinguishing signal responses by eye when using the sensor as a point-of-use diagnostic. We wanted to address this concern in the earlier rounds of the engineering campaign and through ML in case other experimental methods to increase signal responses (i.e., different reporter systems, enriched extract, etc.) were insufficient.

In Round 2B, we observed that most mutants predicted from a ML model trained on normalized DR at zinc and normalized FC at lead had high leak. However, this did suggest that training on DR could impact signal response. Therefore, in Round 3, we trained the model on both (1) normalized FC at lead and zinc and (2) normalized DR at lead and zinc. We screened mutants that appeared in both predictions lists.

Minor

1) In abstract both specificity and selectivity are discussed but not followed in the rest of the paper. What is the difference between specificity and selectivity?

We thank you for pointing this out. We have corrected the text in the abstract to “*sensitivity and selectivity*”.

2) Github site does not seem to be populated?

We appreciate your comment. The GitHub repository was recently updated to improve clarity by refactoring the original utils module into individual, well-documented functions. A Quickstart guide is provided at the bottom of the repository, demonstrating how to apply the functions to a subset of the experimental data. We will ensure that the README is further clarified to make this more immediately visible and user friendly.

3) Details of the plasmids (pJL1) and constructs used in CFP are not sufficient especially details on the promoters and other parts of constructs expressing the TFs. Typically there is a list of plasmids generated at least the initial ones from which the library is based on.

We have now added more details into the Methods section under “DNA library generation” regarding the plasmids used in this work. The DNA sequences for all mutants will also be accessible in the Data Availability files.

4) How reproducible is the FC obtained when the library is screened repeatedly?

We were also concerned about the reproducibility of the FC in the mutants screened, so we ran each screen in duplicate (Round 0) or triplicate (Round 1-3b). In Figure S6, we performed an analysis on the variability in the screening assay between replicates and observed strong correlation between replicates.

Therefore, we are confident that the FC is reproducible. In general, we observe a larger error when using liquid handling robotics (~10-15%) compared to pipetting by hand (5-10%). This motivates us to validate the top variants in by-hand experiments.

5) The line “However, there were two mutants (D64K_N83F and N83I_K104V) that showed a higher fluorescent response to lead than zinc (Fig. S1).” Figure S1 shows D64K_N84F. This must be 83 ? Also Figure S2 x axis labels for round 0 do not seem to correspond to the top two mutants as the N83F mutant seems missing?

We thank you for pointing out the mislabeled mutant in Figure S1. Additionally, the top two mutants can be found in Round 1 of Figure S2. This has been corrected.

6) Typo in the line “We tested high-order mutants because we observed improved sensor activity by combining multiple mutations in our previous PbrR engineering effort” Here the advantage of ML over other random mutations could be mentioned.

We have modified the text as follows:

(Line 289) “These designs were informed directly by prior ML-guided rounds and experimental validation, reflecting a strategic recombination of high performing substitutions. While these rational variants were not proposed de novo by the model, they drew directly from the higher order combinations that the model has already prioritized, which is a targeted exploitation of model discovered signals. Targeted exploitation of high performing mutants through combination has been observed to improve sensor activity by combining multiple mutations in our previous PbrR engineering efforts¹.”

7) References 73 and 74 are not correct. I was expecting to the ProtT5 model and LoRA paper in this line “We use a transformer encoder-decoder model based on the T5 architecture, specifically the ProtT5-XL-UniRef50 model from Rostlab⁷³”

We appreciate the reviewer pointing this out. We have fixed these citations in the revised manuscript.

Reviewer #4 (Remarks on code availability):

GitHub code was empty so could not evaluate the code. Happy to do this in the revised version.

We have now modified the Github so that the code is easily accessible.

References

- 1 Ekas, H. M. *et al.* Engineering a PbrR-Based Biosensor for Cell-Free Detection of Lead at the Legal Limit. *ACS Synthetic Biology* (2024). <https://doi.org/10.1021/acssynbio.4c00456>
- 2 Yang, K. K., Wu, Z. & Arnold, F. H. Machine-learning-guided directed evolution for protein engineering. *Nature Methods* **16**, 687-694 (2019). <https://doi.org/10.1038/s41592-019-0496-6>
- 3 Biswas, S., Khimulya, G., Alley, E. C., Esvelt, K. M. & Church, G. M. Low-N protein engineering with data-efficient deep learning. *Nature Methods* **18** (2021). <https://doi.org/10.1038/s41592-021-01100-y>
- 4 Zhang, Q. *et al.* Integrating protein language models and automatic biofoundry for enhanced protein evolution. *Nature Communications* **16** (2025). <https://doi.org/10.1038/s41467-025-56751-8>
- 5 Yang, J. *et al.* Active learning-assisted directed evolution. *Nature Communications* **16** (2025). <https://doi.org/10.1038/s41467-025-55987-8>
- 6 Huang, C. *et al.* Application of Directed Evolution and Machine Learning to Enhance the Diastereoselectivity of Ketoreductase for Dihydropyridazine Synthesis. *JACS Au* **4** (2024). <https://doi.org/10.1021/jacsau.4c00284>
- 7 Qiu, Y., Hu, J. & Wei, G.-W. Cluster learning-assisted directed evolution. *Nature Computational Science* **1** (2021). <https://doi.org/10.1038/s43588-021-00168-y>
- 8 Vidal, L. S., Isalan, M., Heap, J. T. & Ledesma-Amaro, R. A primer to directed evolution: current methodologies and future directions. *RSC Chemical Biology* **4** (2023). <https://doi.org/10.1039/D2CB00231K>
- 9 Widatalla, T., Rafailov, R. & Hie, B. Aligning protein generative models with experimental fitness via Direct Preference Optimization. *bioRxiv*, 2024.2005.2020.595026 (2024). <https://doi.org/10.1101/2024.05.20.595026>
- 10 Yang, Z. *et al.* Does Negative Sampling Matter? a Review With Insights Into its Theory and Applications. *IEEE Transactions on Pattern Analysis and Machine Intelligence* **46**, 5692-5711 (2024). <https://doi.org/10.1109/TPAMI.2024.3371473>
- 11 Wu, Z., Kan, S. B. J., Lewis, R. D., Wittmann, B. J. & Arnold, F. H. Machine learning-assisted directed protein evolution with combinatorial libraries. *Proceedings of the National Academy of Sciences* **116** (2019). <https://doi.org/10.1073/pnas.1901979116>
- 12 Lobzaev, E., Herrera, M. A., Kasprzyk, M. & Stracquadanio, G. Protein engineering using variational free energy approximation. *Nature Communications* **15** (2024). <https://doi.org/10.1038/s41467-024-54814-w>
- 13 Hie, B., Bryson, B. D. & Berger, B. Leveraging Uncertainty in Machine Learning Accelerates Biological Discovery and Design. *Cell Systems* **11** (2020). <https://doi.org/10.1016/j.cels.2020.09.007>
- 14 Hayes, T. *et al.* Simulating 500 million years of evolution with a language model. *Science* **387** (2025). <https://doi.org/10.1126/science.ads0018>
- 15 Chen, B. *et al.* xTrimoPGLM: unified 100-billion-parameter pretrained transformer for deciphering the language of proteins. *Nature Methods* **2025** (2025). <https://doi.org/10.1038/s41592-025-02636-z>
- 16 Ferruz, N., Schmidt, S. & Höcker, B. ProtGPT2 is a deep unsupervised language model for protein design. *Nature Communications* **13** (2022). <https://doi.org/10.1038/s41467-022-32007-7>
- 17 Nijkamp, E., Ruffolo, J. A., Weinstein, E. N., Naik, N. & Madani, A. ProGen2: Exploring the boundaries of protein language models. *Cell Systems* **14** (2023). <https://doi.org/10.1016/j.cels.2023.10.002>
- 18 Nguyen, E. *et al.* Sequence modeling and design from molecular to genome scale with Evo. *Science* **386** (2024). <https://doi.org/10.1126/science.ado9336>

- 19 Lin, Z. *et al.* Evolutionary-scale prediction of atomic-level protein structure with a language model. *Science* **379** (2023). <https://doi.org/10.1126/science.ade2574>
- 20 Hie, B. L. *et al.* Efficient evolution of human antibodies from general protein language models. *Nature Biotechnology* **42** (2023). <https://doi.org/10.1038/s41587-023-01763-2>
- 21 Yang, J., Li, F.-Z. & Arnold, F. H. Opportunities and Challenges for Machine Learning-Assisted Enzyme Engineering. *ACS Central Science* **10** (2024). <https://doi.org/10.1021/acscentsci.3c01275>
- 22 Saito, Y. *et al.* Machine-Learning-Guided Library Design Cycle for Directed Evolution of Enzymes: The Effects of Training Data Composition on Sequence Space Exploration. *ACS Catalysis* **11** (2021). <https://doi.org/10.1021/acscatal.1c03753>
- 23 Landwehr, G. M. *et al.* Accelerated enzyme engineering by machine-learning guided cell-free expression. *Nature Communications* **16** (2025). <https://doi.org/10.1038/s41467-024-55399-0>
- 24 Ding, K. *et al.* Machine learning-guided co-optimization of fitness and diversity facilitates combinatorial library design in enzyme engineering. *Nature Communications* **15** (2024). <https://doi.org/10.1038/s41467-024-50698-y>
- 25 Kim, G. B., Gao, Y., Palsson, B. O. & Lee, S. Y. DeepTFactor: A deep learning-based tool for the prediction of transcription factors. *Proceedings of the National Academy of Sciences* **118**, e2021171118 (2021). <https://doi.org/doi:10.1073/pnas.2021171118>
- 26 Zeng, W., Dou, Y., Pan, L., Xu, L. & Peng, S. Improving prediction performance of general protein language model by domain-adaptive pretraining on DNA-binding protein. *Nature Communications* **15**, 7838 (2024). <https://doi.org/10.1038/s41467-024-52293-7>
- 27 Watson, J. L. *et al.* De novo design of protein structure and function with RFdiffusion. *Nature* **620** (2023). <https://doi.org/10.1038/s41586-023-06415-8>
- 28 Sumida, K. H. *et al.* Improving Protein Expression, Stability, and Function with ProteinMPNN. *Journal of the American Chemical Society* (2024). <https://doi.org/10.1021/jacs.3c10941>
- 29 Khersonsky, O. *et al.* Automated Design of Efficient and Functionally Diverse Enzyme Repertoires. *Molecular Cell* **72**, 178-186.e175 (2018). <https://doi.org/https://doi.org/10.1016/j.molcel.2018.08.033>
- 30 Padmakumar, V., Pang, R. Y., He, H. & Parikh, A. P. Extrapolative Controlled Sequence Generation via Iterative Refinement. (2023). <https://doi.org/10.48550/arXiv.2303.04562>
- 31 Maret, W. & Li, Y. Coordination Dynamics of Zinc in Proteins. *Chemical Reviews* **109**, 4682-4707 (2009). <https://doi.org/10.1021/cr800556u>
- 32 Cangelosi, V., Ruckthong, L. & Pecoraro, V. L. in *Lead: Its Effects on Environment and Health* (eds Sigel Astrid, Sigel Helmut, & K. O. Sigel Roland) 271-318 (De Gruyter, 2017).
- 33 Aravind, L., Anantharaman, V., Balaji, S., Babu, M. M. & Iyer, L. M. The many faces of the helix-turn-helix domain: Transcription regulation and beyond*. *FEMS Microbiology Reviews* **29**, 231-262 (2005). <https://doi.org/10.1016/j.fmre.2004.12.008>
- 34 Carlson, E. D., Gan, R., Hodgman, C. E. & Jewett, M. C. Cell-free protein synthesis: Applications come of age. *Biotechnology Advances* **30** (2012). <https://doi.org/10.1016/j.biotechadv.2011.09.016>
- 35 Silverman, A. D., Karim, A. S. & Jewett, M. C. Cell-free gene expression: an expanded repertoire of applications. *Nature Reviews Genetics* **21** (2019). <https://doi.org/10.1038/s41576-019-0186-3>
- 36 Hunt, A. C. *et al.* Cell-Free Gene Expression: Methods and Applications. *Chemical Reviews* **125** (2024). <https://doi.org/10.1021/acs.chemrev.4c00116>
- 37 Ekas, H. M. *et al.* An Automated Cell-Free Workflow for Transcription Factor Engineering. *ACS Synthetic Biology* **13** (2024). <https://doi.org/10.1021/acssynbio.4c00471>
- 38 Monchy, S. B. *et al.* Plasmids pMOL28 and pMOL30 of *Cupriavidus metallidurans* Are Specialized in the Maximal Viable Response to Heavy Metals. *Journal of Bacteriology* **189**, 7417-7425 (2007). <https://doi.org/10.1128/jb.00375-07>

- 39 Jarvis, P. & Fawell, J. Lead in drinking water – An ongoing public health concern? *Current Opinion in Environmental Science & Health* **20**, 100239 (2021).
<https://doi.org/10.1016/j.coesh.2021.100239>
- 40 Zietz, B. P., Laß, J., Suchenwirth, R. & Dunkelberg, H. Lead in Drinking Water as a Public Health Challenge. *Environmental Health Perspectives* **118**, a154-a155 (2010).
<https://doi.org/10.1289/ehp.1001979>
- 41 WHO. *Lead poisoning*, <<https://www.who.int/news-room/fact-sheets/detail/lead-poisoning-and-health>> (2024).
- 42 EPA. *Lead Service Lines*, <<https://www.epa.gov/ground-water-and-drinking-water/lead-service-lines>> (2025).
- 43 Oord, A. v. d., Li, Y. & Vinyals, O. Representation Learning with Contrastive Predictive Coding. *arXiv* (2019). <https://doi.org/https://doi.org/10.48550/arXiv.1807.03748>
- 44 Post, M. A Call for Clarity in Reporting BLEU Scores. *arXiv* (2018).
<https://doi.org/10.48550/arxiv.1804.08771>

Reviewer #2 (Remarks to the Author)

The authors have addressed my concerns. The manuscript can be accepted now.

Thank you for your careful consideration of our work and for recommending publication.

Reviewer #3 (Remarks to the Author)

I thank the authors for providing additional explanations as well as additional data. They have mostly addressed my concerns.

We thank you for your time and thoughtful comments on our manuscript.

I would just appreciate if the new increased cross-reactivity with Mercury be discussed-any hints about why this happens only with mercury?

Because the mutations of the final variant are distributed across the different regions of the protein, we hypothesize that both the local changes in the ligand binding domain and the long-range allosteric effects created the new increased cross-reactivity with mercury. Mercury is known to have a high affinity for the thiol groups on cysteines and forms stable Hg-S bonds. Also, mercury prefers a planar trigonal geometry with three cysteines arranged symmetrically¹. PbrR contains three metal-binding cysteines (C114, C79, and C123) that are essential for sensor function^{2,3}. We speculate the mutations created a more favorable coordination geometry (trigonal) for mercury to interact with the three cysteines in the ligand binding domain, resulting in increased cross-reactivity. Additionally, this hypothesis would support the decrease in cross-reactivity to cadmium of the final mutant compared to wildtype as cadmium commonly exhibits tetrahedral coordination geometry when binding to proteins⁴ (similar to zinc).

We have modified Line 435 as follows:

“The mutations we identified may subtly reshape the binding pocket’s geometry or electrostatics to disfavor zinc coordination and improve lead coordination, as well as increase cross reactivity to mercury (Fig. S4).”

I agree that given the different LOD, the sensor is still relevant for lead, but would like a bit more discussion on the potential impacts and implication of this new cross-reactivity generated by the workflow. For example: are they found commonly together in drinking water? I seems not from a brief survey. In what particular set ups this cross reactivity would be a problem?

Zinc and lead can be found together in drinking water due to corrosion from galvanized pipes. For example, one of the municipal water samples we used in Figure 6 contained 0.13 μM lead and 5.2 μM zinc (Source Data file).

The main reason we set out to create a lead biosensor with high lead sensitivity and no zinc sensitivity is to eliminate the occurrence of false positives. Wildtype PbrR exhibited strong

sensitivity to zinc concentrations that are below the EPA maximum limit (Fig. 2A). If a wildtype PbrR-based biosensor was used to test a water sample containing zinc but no lead, it is possible that it would turn on, resulting in a false positive. By engineering PbrR for lead selectivity over zinc, we significantly reduce the likelihood of this false positive.

We have added another comment about avoiding false positives to the discussion:

“Furthermore, we showed a 500-fold improvement in sensitivity from a previous report⁵, while avoiding Zn selectivity problems that have plagued past efforts with false positives⁶.”

Reviewer #5 (Remarks to the Author):

I reviewed the revised manuscript and responses to previous reviewers and I believe the manuscript is adequately revised and ready for publication.

We appreciate your effort in evaluating our manuscript and recommending publication.

References

- 1 Wang, D. *et al.* Structural Analysis of the Hg(II)-Regulatory Protein Tn501 MerR from *Pseudomonas aeruginosa*. *Scientific Reports* 2016 6:1 **6** (2016-09-19). <https://doi.org/10.1038/srep33391>
- 2 Hobman, J. L. *et al.* Cysteine coordination of Pb(II) is involved in the PbrR-dependent activation of the lead-resistance promoter, PpbrA, from *Cupriavidus metallidurans* CH34. *BMC Microbiology* 2012 12:1 **12** (2012-06-18). <https://doi.org/10.1186/1471-2180-12-109>
- 3 Tolbatov, I., Re, N., Coletti, C. & Marrone, A. Determinants of the Lead(II) Affinity in pbrR Protein: A Computational Study. *Inorganic Chemistry* **59** (December 12, 2019). <https://doi.org/10.1021/acs.inorgchem.9b03059>
- 4 Tebo, A. G., Hemmingsen, L. & Pecoraro, V. L. Variable primary coordination environments of Cd(II) binding to three helix bundles provide a pathway for rapid metal exchange. *Metallomics* **7**, 1555-1561 (2015). <https://doi.org/10.1039/c5mt00228a>
- 5 Jung, J. K. *et al.* Cell-free biosensors for rapid detection of water contaminants. *Nature Biotechnology* 2020 38:12 **38** (2020). <https://doi.org/10.1038/s41587-020-0571-7>
- 6 Ekas, H. M. *et al.* Engineering a PbrR-Based Biosensor for Cell-Free Detection of Lead at the Legal Limit. *ACS Synthetic Biology* (2024). <https://doi.org/10.1021/acssynbio.4c00456>